# Plant community stability is associated with a decoupling of prokaryote and fungal soil networks

Dina in 't Zandt ®[1] ✉, Zuzana Kolaříková[1], Tomáš Cajthaml[2,3] &
Zuzana Münzbergová ®[1,4]

Soil microbial networks play a crucial role in plant community stability. However, we lack knowledge on the network topologies associated with stability and the pathways shaping these networks. In a 13-year mesocosm experiment, we determined links between plant community stability and soil microbial networks. We found that plant communities on soil abandoned from agricultural practices 60 years prior to the experiment promoted destabilising properties and were associated with coupled prokaryote and fungal soil networks. This coupling was mediated by strong interactions of plants and microbiota with soil resource cycling. Conversely, plant communities on natural grassland soil exhibited a high stability, which was associated with decoupled prokaryote and fungal soil networks. This decoupling was mediated by a large variety of past plant community pathways shaping especially fungal networks. We conclude that plant community stability is associated with a decoupling of prokaryote and fungal soil networks and mediated by plant-soil interactions.

Plants associate with complex interactive networks of soil microbial communities. These networks are increasingly found to act as a structuring force in plant community stability processes[1–5]. Plant community stability describes the ability of communities to resist and recover from biotic and abiotic perturbations, and has become an increasingly pressing issue with the ongoing change in climate and human interventions in natural ecosystems[6,7]. However, understanding the driving forces of community stability is a major challenge due to the complexity of the underlying plant-soil-microbiota interactions. At the same time, the complexity of ecological interactions itself has long been considered to be a key component of community stability[8,9]. To predict, protect and restore plant communities, we need to understand the role of plant-soil-microbiota interactions in community stability processes.

Ecological interaction networks have a coherent structure with well-defined patterns in relation to network stability[8,9]. Network stability results from species connectiveness, negative interactions, few strong and many weak interactions and the clustering of species into subgroups[7,10–12]. Clustering involves many connections within subgroups of species and fewer connections between them, effectively decoupling the subgroups within the community[13]. In essence, these network properties minimise the risk of change when a perturbation occurs by creating dependencies between species, promoting species asymmetry and buffering against the propagation of perturbation effects among subsections of the network[10,12–16].

Ecological network theory is derived from ecological food web theory and network science, but has been shown to be applicable to microbial networks[7,10,14]. However, in comparison to food webs and many other networks, microbial networks lack a strong directional structure and are based on co-occurrences of taxa alone[10]. Yet, we know that plants play a critical role in shaping the environmental niches of soil microbial communities via the input of a large variety of chemical

[1]Institute of Botany, Czech Academy of Sciences, 252 43 Průhonice, Czech Republic. [2]Institute for Environmental Studies, Faculty of Science, Charles University, Praha 2, Czech Republic. [3]Institute of Microbiology, Czech Academy of Sciences, Vídeňská 1083, Prague CZ-14220, Czech Republic. [4]Department of Botany, Faculty of Science, Charles University, Praha 2, Czech Republic. ✉e-mail: dina.intzandt@ibot.cas.cz

compounds into the soil environment by, for example, the input of dead organic material and root exudates as well as the uptake of soil nutrients[17,18]. The plant community may therefore play an essential role in shaping soil microbial network stability. We currently lack knowledge on the role of associations between the plant community and soil microbial networks in enhancing whole community stability.

Generally, interactions between plants and soil microbiota are increasingly found to act as a stabilising force in plant community processes[2–5]. Plant species identity and thus plant community composition is expected to determine the potential for stabilising mechanisms via reciprocal specialisation of plants and soil microbiota. This reciprocal specialisation creates complex networks and, in particular, negative feedback loops that avoid plant species dominance, species loss and communities tipping into alternative states[2–5]. Direct interactions between plant community composition and microbial networks are therefore likely critical pathways in community stabilising mechanisms. Conversely, inherently more generic interactions of indirect plant community pathways mediated by soil chemical changes may result in microbiota responding in tandem. If such more generic effects are strong or not compensated for, these in tandem responses may lead to plant community instability. To understand the drivers of plant community stability, we need to define the importance of both overall (i.e., plant diversity and productivity) and compositional (where plant identity plays a distinctive role) plant community interactions with soil microbial networks and test whether these plant community components associate directly or indirectly via soil chemical changes.

Here, we test to what extent and via which pathways overall and compositional plant community components associate with soil microbial biomass and networks. We compare these pathways between dry grassland communities established on natural grassland soil and soil abandoned from agricultural practices 60 years before the start of the experiment. The two soils mainly differed in soil nutrient availability with the natural grassland soil being significantly lower in total N, organic C and plant available P and K than the abandoned arable soil[19]. Plant communities on natural grassland soils are typically stable communities, while communities on abandoned arable soil are more strongly impacted by invading plant species destabilising plant community networks[20,21]. We created diverse plant communities by sowing a seed mixture of 44 perennial dry grassland species in outdoor mesocosms filled with natural grassland soil and abandoned arable soil[19]. Plant communities were left to establish for 5 years, after which natural invasion by both native and exotic species from outside the sown species pool was allowed and occurred substantially the following 8 years. Long-term plant community development resulted in communities with natural variation in, amongst others, plant invasion impact and plant community composition. We combined four datasets: plant community aboveground measurements over the 13 years and soil chemistry, total microbial biomass (PLFA/NLFA analysis) and soil microbial community composition (16S and ITS amplicon sequencing) after the 13th growing season.

First, we test whether plant communities on abandoned arable soil show a decreased long-term stability aboveground and whether this translates to soil microbial communities with destabilising properties in their prokaryote and fungal co-occurrence networks. Second, using structural equation modelling (SEM), we test whether the relative contribution of direct and indirect pathways of the overall plant community (aboveground productivity and plant diversity) and plant composition onto soil microbial networks is affected by soil origin. We distinguish past plant community factors (initial plant invasion impact and developmental trajectories) from factors in the year of sampling as well as direct plant-microbial pathways from indirect pathways occurring via soil chemical changes. Third, we determine the most important plant-soil-microbiota pathways that are consistently changed between stable and instable plant communities, and whether these changes relate to particular putative functions and metabolic

characteristics of the microbial communities involved. Taken together, these analyses unfold the pathways via which plant communities associated with soil microbial networks and the role of plant-soil-microbial interactions in plant community stability.

## Results

### Natural grassland communities had a lower plant invasion and higher temporal stability

Plant community diversity gradually declined in time, but was drastically increased after the start of plant invasion on both natural grassland and abandoned arable soil (Fig. 1a). Despite this variation in diversity, aboveground productivity of the communities remained relatively constant over time. In addition, aboveground productivity was, on average, not affected by the start of invasion and showed little difference between the two soil origins (Fig. 1b). The proportion of invaded species biomass, on the other hand, increased with the onset of invasion and was consistently higher in abandoned arable soil communities than in communities established on natural grassland soil (Fig. 1c). For each community, we calculated stability over time and found that plant communities on natural grassland soil had a significantly higher temporal stability than plant communities on abandoned arable soil (Fig. 1d).

Plant community composition in time was analysed using detrended correspondence analysis (DCA). This analysis described variation in community composition by three axes: variation relating to the gain and loss of species over time (temporal turnover separating early from late residency species; DCA 1), variation relating to Ellenberg indicator values of the plant species' soil resource optima (separating species with high and low soil resource optima; DCA 2), and variation relating to legume cover (separating communities with high and low legume cover; DCA 3) (Fig. 1e–g, Supplementary Fig. 1). DCA 1 showed a relative similar turnover in plant community composition over time between the two soil origins. In addition, on both soils a sharp drop in early residency species occurred with the start of plant invasion in 2012 (Fig. 1e, Supplementary Fig. 1a). For DCA 2, the start of plant invasion marked a strong increase in plant species with high soil resource optima as most invading species were characterised by having a high ecological soil resource optimum (Fig. 1f, Supplementary Fig. 1a). This increase was strongest for communities on abandoned arable soil and persisted in the following years (Fig. 1f, Supplementary Fig. 1a). Finally, DCA 3 separated plant community composition between the two soil origins before the start of invasion: communities on natural grassland soil harboured a higher cover of legumes than communities on abandoned arable soil in the years before 2012. With the start of invasion, plant community composition on natural grassland soil dropped to, on average, similar legume cover as for the abandoned arable communities (Fig. 1g, Supplementary Fig. 1b).

### Soil origin affected soil chemistry and microbial soil communities

After the 13th growing season, natural grassland and abandoned arable soils differed significantly in chemistry, microbial biomass and microbial community composition (Fig. 2a, b, Supplementary Fig. 2). Abandoned arable soil was significantly higher in total N, plant available $NO_3^-$, $NO_2^-$, and $NH_4^+$, total and organic C and plant available P (Supplementary Fig. 2). In addition, prokaryote and fungal richness as well as bacterial biomass were significantly higher in abandoned arable than natural grassland soil (Supplementary Fig. 2). Conversely, soil pH, fungal and AMF biomass were higher in natural grassland soil (Supplementary Fig. 2). Both prokaryote and fungal community composition were significantly different between natural grassland and abandoned arable soil (Fig. 2a, b). These differences largely resulted from differences in prokaryote and fungal OTU abundances given the large overlap in the OTUs that were present in both soils (Supplementary Fig. 3a, b).

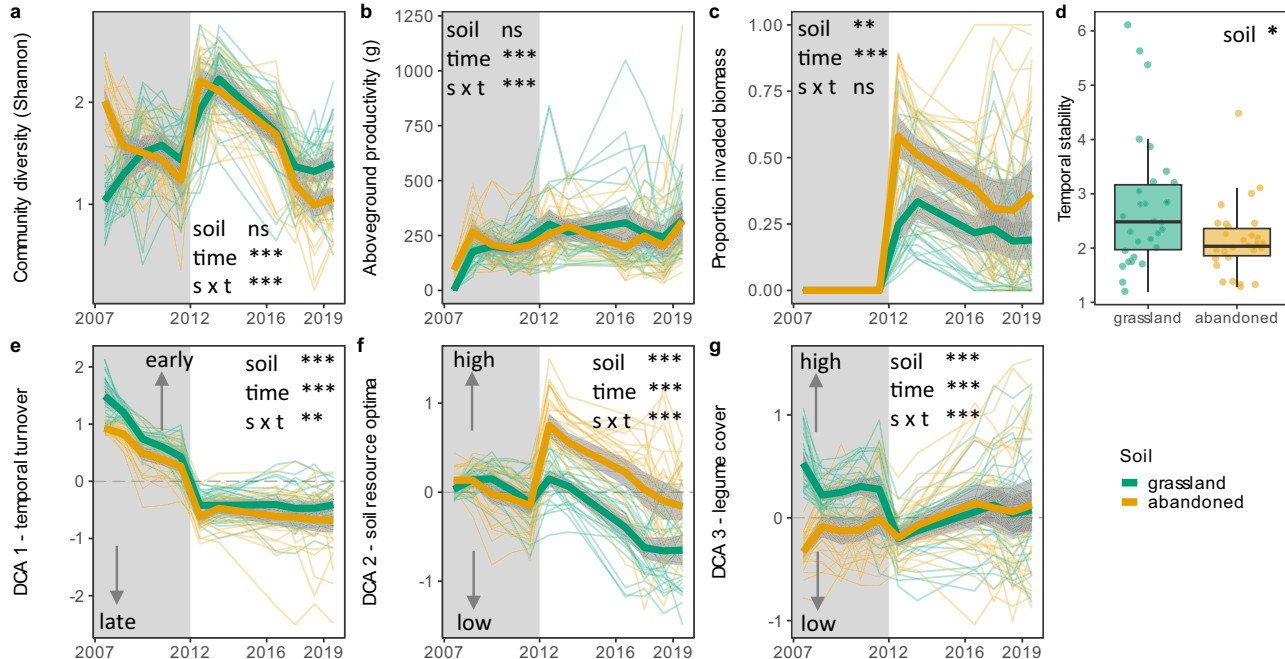

**Fig. 1 | Plant community development and temporal stability.** Plant community (**a**) diversity, (**b**) aboveground productivity, (**c**) biomass proportion of invaded plant species, (**d**) temporal stability based on aboveground productivity, and community compositional DCA scores related to plant species (**e**) temporal turnover, (**f**) ecological soil resource optimum and (**g**) legume cover of communities grown on natural grassland (green) and abandoned arable soil (yellow). In (**e**–**g**) arrows with text indicate the interpretation of the DCA scores. Grey shading indicates the time period in which communities established and no natural species invasion took place. From 2012 onwards, natural invasion of species from outside the sown species pool occurred. Thick lines indicate averages with the shading showing the 95% confidence intervals. Thin lines show each replicate ($n = 30$). Temporal stability was calculated as the inverse coefficient of variation of aboveground productivity over time. Boxplots indicate median (middle line), 25th, 75th percentile (box) and 5th and 95th percentile (whiskers) with single points indicating each replicate. Results of linear mixed effect models including sowing density as a random effect and type III Wald chi-square tests are presented. Significance codes: ***$p < 0.001$; **$p < 0.01$; ns = not significant, $p > 0.05$. For figures on species distribution on the DCA axes, see Supplementary Fig. 1. For exact statistical values, see Supplementary Data 1. Source data are provided as a Source Data file.

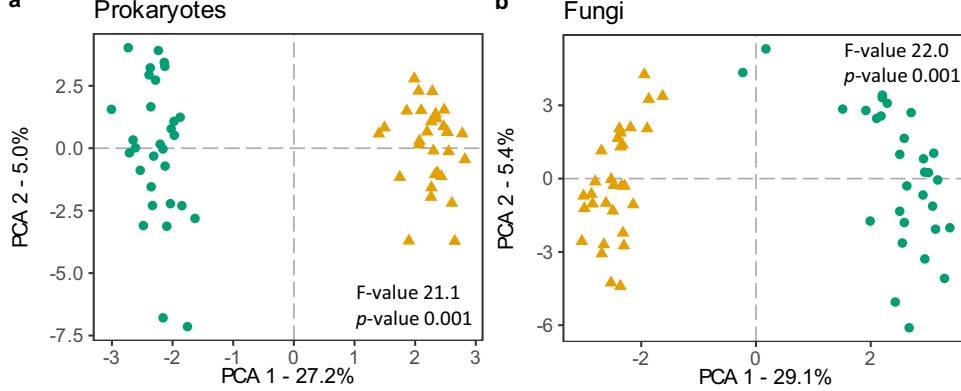

**Fig. 2 | Principal component analysis (PCA) of soil prokaryote and fungal communities.** Soil (**a**) prokaryote 16S and (**b**) fungal ITS rRNA reads in plant communities established on natural grassland (green circles) and abandoned arable soil (yellow triangles). Amplicon sequencing was performed at the end of the 13th growing season ($n = 30$ per soil origin). PCA analysis was performed on clr-transformed read counts. Significant separation of the communities between the two soils was tested using permutational multivariate analysis of variance (PERMANOVA) based on Euclidean distances. Source data are provided as a Source Data file.

## Temporal stability was associated with a decoupling of prokaryote and fungal networks

For each soil origin, we created microbial co-occurrence networks. The networks indicated highly connected prokaryote and fungal communities in both natural and abandoned soil with few dominating OTUs and little difference in network properties commonly associated with network stability (Supplementary Fig. 4; Supplementary Table 1). To understand the microbial network topologies in more detail, we clustered similarly responding OTUs across the 30 plant communities of each soil origin (Supplementary Fig. 4). Similarly responding prokaryote OTUs were captured in 9 and 10 clusters for natural grassland and abandoned arable soil, respectively. For fungi, 21 and 18 clusters were needed for natural grassland and abandoned arable soil communities, respectively (Fig. 3a, b). In all microbial networks, taxonomic families largely clustered together, indicating similar responses of closely related taxa (Supplementary Figs. 5–6; Supplementary Tables 2–5). Importantly, all microbial networks showed a significantly denser clustering of OTUs than would be expected based on null-models (randomly rewired networks; Supplementary Fig. 7), indicating a distinct organisational pattern of each network.

**a**  Natural grassland

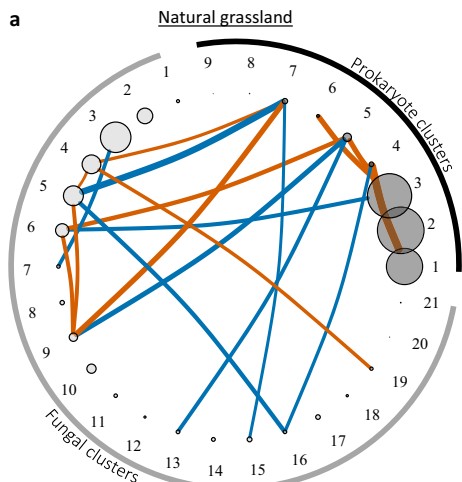

**b**  Abandoned arable

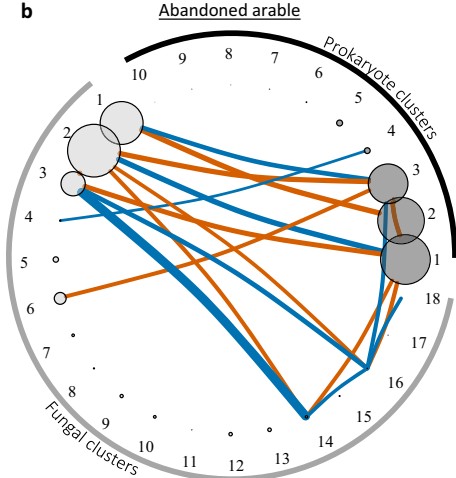

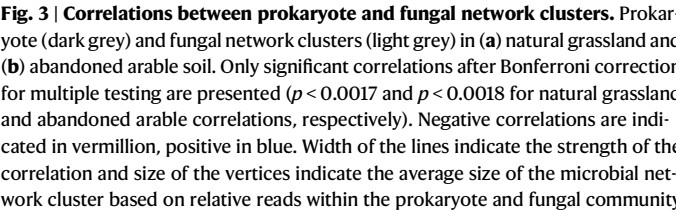

**Fig. 3 | Correlations between prokaryote and fungal network clusters.** Prokaryote (dark grey) and fungal network clusters (light grey) in (**a**) natural grassland and (**b**) abandoned arable soil. Only significant correlations after Bonferroni correction for multiple testing are presented ($p < 0.0017$ and $p < 0.0018$ for natural grassland and abandoned arable correlations, respectively). Negative correlations are indicated in vermillion, positive in blue. Width of the lines indicate the strength of the correlation and size of the vertices indicate the average size of the microbial network cluster based on relative reads within the prokaryote and fungal community each. Note that this means that cluster sizes can be compared between the two soils within the prokaryotes and fungal groups, but that prokaryote and fungal clusters are not scaled to each other and are therefore not directly comparable (but see Supplementary Fig. 2 for bacterial and fungal biomass comparison). For taxonomic and putative soil functions of each microbial network cluster see Supplementary Figs. 5–6 and Supplementary Tables 2–5. For exact statistical values, see Supplementary Data 1. Source data are provided as a Source Data file.

We summed the relative abundance of the OTUs in each cluster. On average, three dominant prokaryote clusters occurred that held, on average, 86% and 91% of the 16S rRNA reads recovered in natural grassland and abandoned arable soil, respectively (Fig. 3a, b, Supplementary Fig. 5a, b). In abandoned arable soil, these patterns were similar for the fungal networks showing three large clusters holding, on average, 75% of the ITS rRNA reads. On natural grassland soil, however, fungal networks showed five larger clusters holding, on average, 62% of the ITS rRNA reads (Fig. 3a, b, Supplementary Fig. 5c, d).

We tested for significant correlations between all clusters in each soil origin. Most strikingly, we observed that in natural grassland soil, only one significant correlation between the dominant prokaryote and fungal clusters was present, specifically between prokaryote cluster 3 and fungal cluster 6 (Fig. 3a). Conversely, in abandoned arable soil, several strong positive and negative correlations occurred between the dominant prokaryote and fungal clusters, specifically between prokaryote clusters 1–3 and fungal clusters 1–3 (Fig. 3b).These findings suggest that responses of dominant prokaryote and fungal clusters in natural grassland soil were decoupled, while in abandoned arable soil, these clusters responded in tandem. Importantly, these patterns were not random and occurred significantly more often than expected by chance in both soil origins, highlighting the distinct organisational structure of the prokaryotes and fungi in the microbial networks (Supplementary Fig. 8).

For each plant community, we then calculated the strength of coupling between prokaryote and fungal network clusters. We found that the coupling of prokaryote and fungal clusters was significantly related to aboveground temporal stability: temporal stability was associated with a decoupling of prokaryote and fungal network clusters (Wald test, $\chi^2 = 8.817$, $p = 0.003$; Fig. 4).

**Microbial communities in abandoned arable soil were predominantly associated with the plant community in the year of sampling**

We hypothesised that the plant community plays an important role in shaping soil microbial networks and biomass, and may underlie the coupling/decoupling of prokaryote and fungal networks. We tested this hypothesis using structural equation modelling (SEM). For each soil origin, we determined how plant community parameters from the year of sampling and from the past (initial plant invasion impact in 2012 and plant community developmental trajectories) associated with microbial biomass and network clusters at the end of the 13th growing season (Fig. 5a). We tested whether these associations resulted from overall plant community factors (aboveground productivity and plant diversity) or plant community composition (DCA scores), and determined whether these pathways occurred via direct or indirectly pathways via soil chemical changes (Fig. 5a). The obtained SEM models explained, on average, 47% of the variation in microbial biomass. For microbial clusters, the explained variation varied strongly between the network clusters in both soil origins and lay between 0 and 64% (Supplementary Table 6).

We calculated the relative contribution of each group of plant community parameters in shaping soil microbial properties using the effect sizes of the SEM pathways. Most strikingly, we found that plant community parameters in the year of sampling played a much larger role in shaping both microbial biomass and networks in abandoned arable than natural grassland soil (Fig. 5b).

Furthermore, the main players in the year of sampling in abandoned arable soil were direct associations with the overall plant community and indirect associations of plant composition. In contrast, in natural grassland soil, associations in the year of sampling most consistently occurred via direct plant compositional pathways. In addition, a few indirect plant compositional pathways occurred for prokaryote and fungal network clusters, but not for microbial biomass. Direct associations with the overall plant community occurred solely for fungal clusters (Fig. 5b).

A variety of past plant community parameters were important in shaping microbial communities in natural grassland soil. Microbial biomass was largely shaped by direct and indirect pathways of past overall plant community factors. Prokaryote and fungal clusters were largely shaped by indirect pathways of the past overall plant community as well as past direct compositional pathways (Fig. 5b). In abandoned arable soil, most of these past pathways were absent with exception of direct compositional effects on prokaryote network clusters (Fig. 5b).

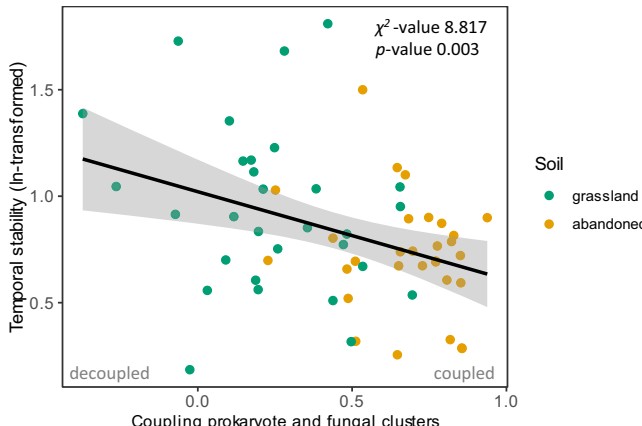

**Fig. 4 | Temporal stability and the coupling of prokaryote and fungal networks.** Correlation of temporal stability aboveground (ln-transformed) and the coupling of prokaryote and fungal clusters in soil microbial networks in natural grassland (green) and abandoned arable soil (yellow). Temporal stability was calculated as the inverse coefficient of variation in aboveground productivity of each plant community (Fig. 1d). The coupling of prokaryote and fungal responses was estimated by the slope (beta) of the relative abundance of prokaryotes and fungi in network clusters 1–9 (Fig. 3). The x-axis shows the beta slope coefficient. A positive value indicates that prokaryote and fungal reads were distributed similarly over the most prevalent network clusters, and thus, that prokaryote and fungal responses were coupled. A value of zero indicates that prokaryote and fungal read distribution over the network clusters was unrelated and thus decoupled. A few weak, negative correlations occurred and indicate that prokaryote and fungal read distribution over the network clusters were nearing an opposite pattern, indicating a strong decoupling of prokaryote and fungal responses. Results from a linear mixed effects model including soil as a random factor and type III Wald chi-square test are shown ($n = 60$). The black line indicates the average correlation with the shading showing the 95% confidence interval. Source data are provided as a Source Data file.

## Prokaryote and fungal networks were associated with unique past pathways in natural grassland soil

To understand the mechanisms that underlie the association between stability and the coupling/decoupling of prokaryote and fungal soil networks, we compared the specific processes associated with prokaryote and fungal network clusters in natural grassland and abandoned arable soil. For this we extracted all direct and indirect plant community pathways with a relative contribution > 5% and determined whether these pathways occurred for both prokaryotes and fungal clusters or were unique to either group (Fig. 6a, b).

In natural grassland soil, both prokaryote and fungal clusters were for an important part shaped by unique pathways. These resulted especially from past plant community parameters (Fig. 6a). For natural grassland prokaryote clusters, unique pathways resulted from plant compositional trajectories in time and, to a lesser extent, the initial effect size of plant invasion in 2012. Similarly, unique pathways for natural grassland fungal clusters resulted from plant compositional trajectories in time and the effect size of plant invasion, but different sets of plant compositional axes were involved. In addition, for fungal clusters, changes in soil pH and organic C played an important role in past pathways (Fig. 6a).

In abandoned arable soil, less differentiation in past pathways between prokaryote and fungal networks occurred (Fig. 6b). Some similarities to the natural grassland soil occurred, such as unique pathways for prokaryote and fungal clusters resulting from the initial effect size of invasion in 2012 (Fig. 6b). However, on abandoned arable soil, overlapping pathways associating with plant diversity and composition in the year of sampling were more numerous than unique past pathways (Fig. 5b).

## In the year of sampling, plant composition decreased fungal community specialisation in abandoned arable soil

As a final step, we searched for generalities in the prokaryote and fungal characteristics that were associated with plant community pathways by testing which pathways significantly associated with overall microbial community patterns: alpha-diversity (Shannon's index) and the relative habitat specialisation index (SI). In addition, we sought to understand the functional traits of each microbial network cluster by analysing their taxonomic composition and comparing it to existing literature and databases, and calculating the taxa's relative habitat specialisation and enrichment in natural grassland or abandoned arable soil (Supplementary Figs. 9–10; Supplementary Tables 2–5).

In the year of sampling, we found that plant composition did not affect any of the soil nutrients in natural grassland soil, only belowground productivity was affected (Fig. 6a; Supplementary Table 7). As a result, most plant compositional pathways in the year of sampling occurred via direct interactions. A high legume cover in the year of sampling was associated with a low fungal and AMF biomass, and plant communities with a high cover of high resource optima plant species were associated with a low fungal diversity and bacterial biomass. In the latter case, belowground productivity was low and therewith increased prokaryote diversity (Supplementary Table 7). Furthermore, in contrast to abandoned arable soil, plant diversity did not have detrimental effects on the fungal community and rather increased fungal diversity in natural grassland soil (Supplementary Table 7). Most of the pathways in natural grassland soil involved metabolically diverse groups of bacteria, putative soil and litter saprotrophs as well as plant pathogens (Supplementary Table 7).

In abandoned arable soil, we found that plant diversity and composition in the year of sampling associated with a large number of prokaryote and fungal network clusters. In contrast to natural grassland soil, the plant compositional pathways occurred via changes in soil pH, total N and $NO_3^-$ in abandoned arable soil (Fig. 6b). Most of these pathways involved dominant prokaryote and fungal clusters, putative soil saprotrophs, plant pathogens, ammonia oxidising archaea (AOA) and metabolically diverse bacteria (Supplementary Table 8). Interestingly, a high cover of high resource optima plant species and a low cover of legumes resulted in a low fungal community specialisation, which was mediated through an increase in soil pH (Supplementary Table 8). In addition, a high legume cover and high plant diversity decreased fungal biomass (Supplementary Table 8; Supplementary Figs. 11b–12b).

## Past plant community pathways were especially associated with putative saprotrophs and pathogens

In natural grassland soil, past plant community pathways were numerous and diverse, but for an important part related to the initial impact that plant species invasion had on the plant communities (Supplementary Table 7; Supplementary Fig. 11). A large increase in aboveground productivity with the start of plant invasion in 2012 was associated with an increased soil N and decreased soil pH after the 13th growing season (2019). Subsequently, fungal and bacterial biomass as well as prokaryote community specialisation were increased. Similarly, a large increase in plant diversity with invasion in 2012 was associated with a decreased soil organic C (Supplementary Table 7; Supplementary Fig. 12a). These past plant community pathways involved a large diversity of microbiota, but especially diverse groups of chemoheterotrophic bacteria, putative soil saprotrophs and plant pathogens (Supplementary Table 7).

Importantly, in natural grassland soil, large changes in plant composition with the start of plant invasion affected many fungal, but not prokaryote clusters. Prokaryote clusters were instead associated with plant compositional trajectories over time (Fig. 6a; Supplementary

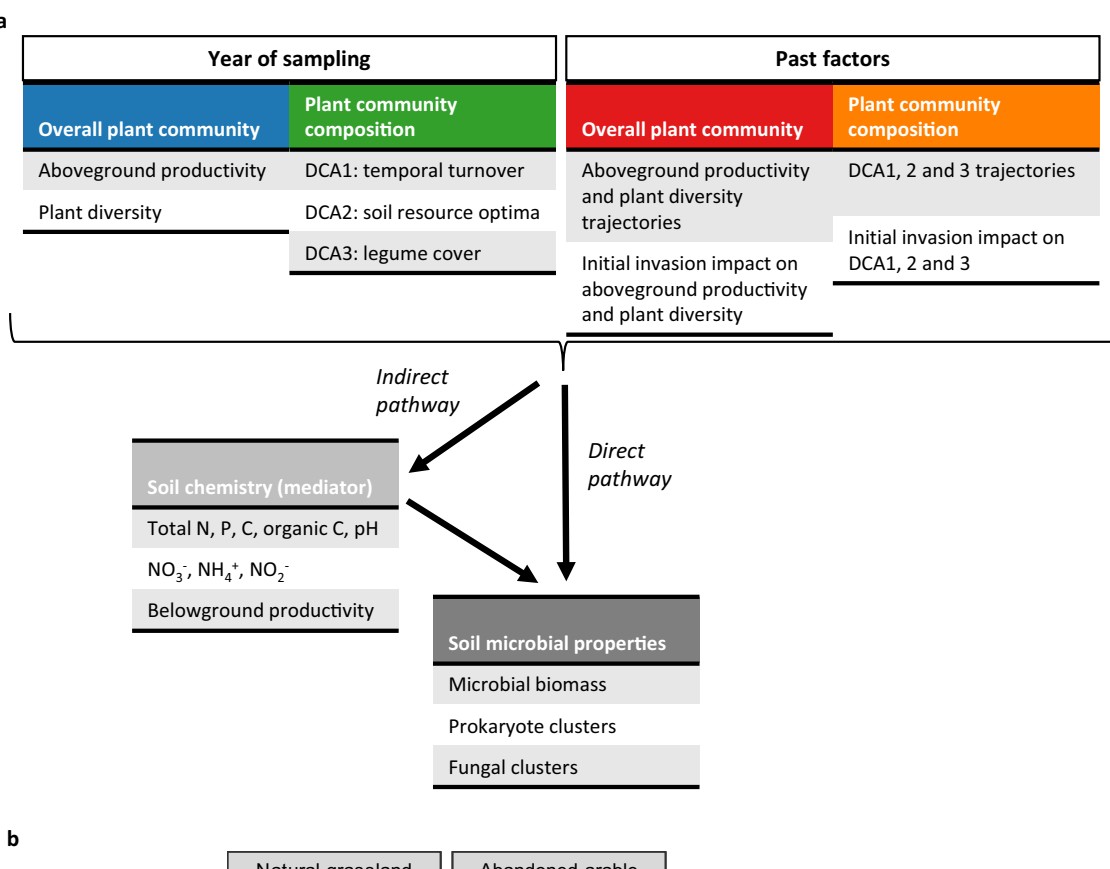

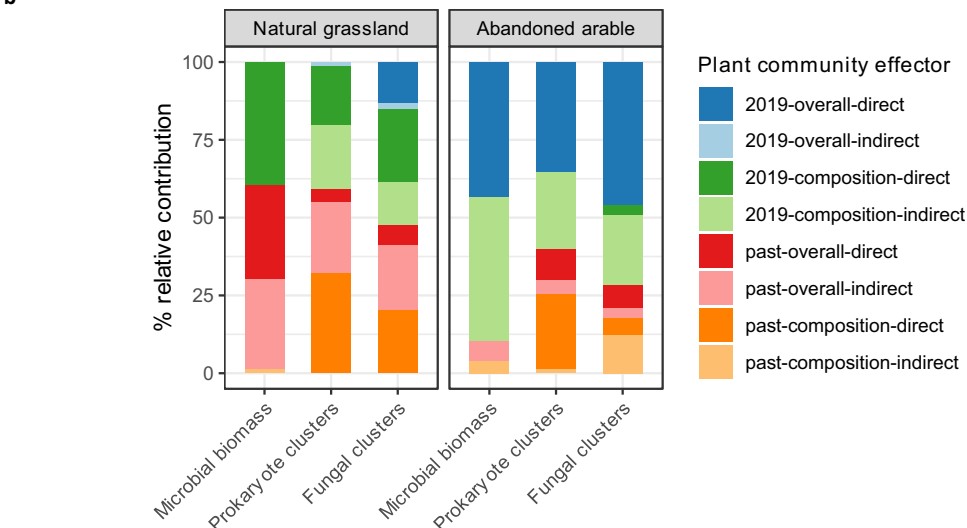

**Fig. 5 | Structural equation modelling (SEM). a** Plant community factors implemented in the SEM approach from the year of sampling (2019) and the past (2012–2019), the considered soil chemical factors as mediators in indirect pathways and the soil microbial properties that were associated either via direct or indirect pathways (*n* = 30). **b** Combined, relative contribution of each group of plant community factors in shaping soil microbial properties in natural grassland and abandoned arable soil. The relative contribution is based on the effect sizes of the SEM pathways weighted by the size of the involved microbial parameters (biomass or the average relative size of the network cluster). Values were corrected for the number of potential pathways in the SEM to allow for direct comparison of the various microbial properties. Colours between a and b match the four main groups of plant factors considered. In (**b**), dark colours indicate direct pathways and faded colours indirect pathways. Note that the strict directionality of the indirect pathway is a mathematical necessity to test whether soil chemistry is a significant mediator, but should be interpreted as an interactions as both plants and microbiota influence soil chemistry. For information on the explicit pathways, see Supplementary Figs. 11–12. Source data are provided as a Source Data file.

Table 7). The various fungal clusters affected by plant invasion impact especially consisted of putative soil saprotroph and plant pathogens (Supplementary Table 7). For this latter group, plant communities where few early plant species were lost with the onset of plant invasion were increased in a specific set of putative soil pathogens. In addition, communities with a large increase in plant species with high soil resource optima were increased in a set of generalist soil pathogens, but decreased in another set of putative pathogens (Supplementary Table 7).

In abandoned arable soil, past plant community pathways also occurred for an important part via the initial impact of plant species invasion in 2012. These pathways, however, consistently affected both

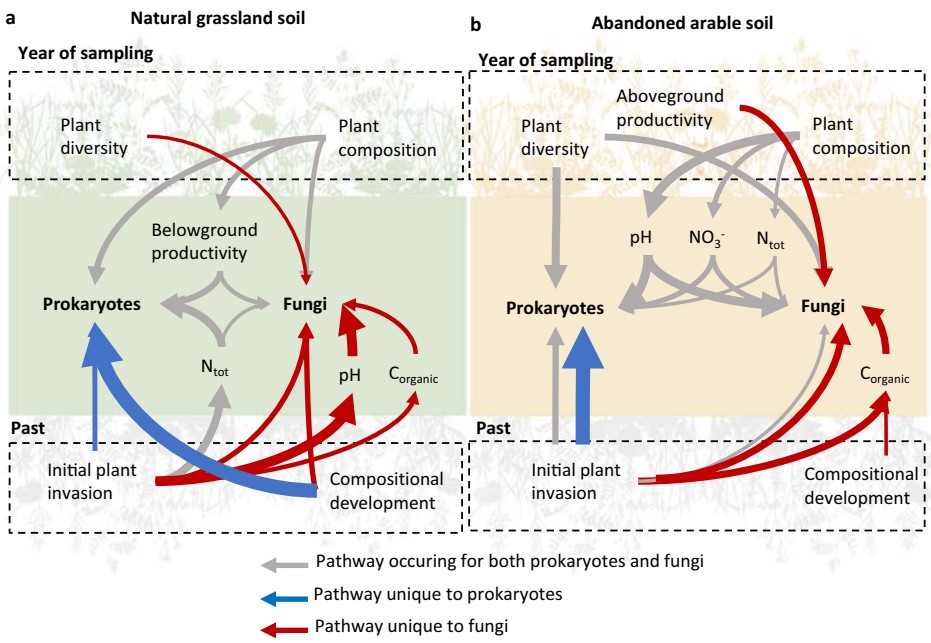

**Fig. 6 | Summary of the most important SEM pathways.** Plant community and soil chemical factors shaping prokaryote and fungal soil network clusters in (**a**) natural grassland soil and (**b**) abandoned arable soil. Grey arrows indicate pathways occurring for both prokaryote and fungal clusters, blue arrows pathways unique to prokaryote clusters and red arrows pathways unique to fungal clusters. Arrow width indicates the relative contribution of the pathway. Pathways related to plant community parameters in the year of sampling (2019; top part of figure) and from the past (2012–2019; bottom part of figures) were considered. The most important pathways were defined based on the relative contribution of plant community parameters (>5% contribution): the effect size of the pathway weighted by the relative cluster size (Fig. 3). Pathways are grouped and therefore multiple arrows can occur if, for example, different compositional axes are involved for prokaryotes and fungi. For exact statistical values, see Supplementary Data 1.

prokaryote and fungal clusters and generally affected especially diverse groups of prokaryotes as well as putative soil saprotrophs and plant pathogens (Fig. 6b; Supplementary Table 8). Interestingly, almost all affected plant pathogens occurred in a single, dominant cluster, which was mediated by changes in soil organic C (Supplementary Table 8).

## Discussion
### Plant community stability is associated with a decoupling in prokaryote and fungal responses
We tested how plant community stability associated with soil microbial networks. We found that above ground stability was associated to a decoupling of prokaryote and fungal responses below ground. Soil abandoned from agricultural practices 60 years before the start of the 13-year long mesocosm experiment had destabilising effects on dry grassland community plant-soil-microbial networks, compared to natural grassland soil. This instability was demonstrated by a lower aboveground temporal stability and a higher success of invading plants in communities grown on abandoned arable soil, which is in line with previous studies[20,21]. Furthermore, in abandoned arable soil, the dominant fungal and prokaryote network clusters had strong positive and negative co-occurrences, indicating a joint response to environmental factors. This response was tied to the plant community, as both the prokaryote and fungal clusters were associated with relatively similar pathways related to plant-soil interactions. These patterns are characteristic for instable networks, as perturbation effects can easily spread throughout the network, affecting large proportions of a network rather than just a small section (Fig. 7)[11,22]. This highlights the sensitivity of the abandoned arable microbial networks to disturbances and the potential for major restructuring.

In contrast, we found that prokaryote and fungal responses in natural grassland soil were largely decoupled. The prokaryote and fungal networks clusters were, for an important part, associated to separated pathways related to the plant community. This division

likely created separate soil niches for each group[22,23], leading to enhanced resilience against disturbances in the system. The decoupled structure of microbial networks is associated with the ability to buffer against the spread of disturbances: a disturbance affecting one or a few of the microbial clusters will not cascade into affecting unconnected clusters (Fig. 7; compartmentalization[11,22]). We conclude that plant community stability is associated with a decoupling of prokaryote and fungal responses, which is likely shaped by interactions with the plant community.

### Plant-mediated soil legacies play a key role in decoupling microbial networks
We show that the soil microbial networks in natural grassland soil were more strongly associated with past plant community pathways than the networks in abandoned arable soil. These past plant community pathways may have separated the responses of prokaryote and fungal network clusters in natural grassland soil. This suggests that plant-mediated soil legacies play a crucial role in decoupling fungal and prokaryote responses, resulting in a more stable soil microbial community (Fig. 7).

Plant-mediated soil legacies are well-known phenomena and play a driving role in plant community coexistence, diversity and succession[2–4]. We found that plant-mediated soil legacies resulted, for an important part, from the initial impact of plant species invasion in 2012. In natural grassland soil, drastic increases in plant diversity, aboveground productivity and changes in plant composition resulted in numerous changes in putative fungal saprotrophs after the 13th growing season. Together with the lasting impact of the initial plant invasion on soil organic C, pH, and total N, our findings suggest that past invasion pathways are largely due to changes in decomposition processes. This is consistent with the observed association of these pathways with fungal networks only, as fungi are well-known for their crucial role in decomposing plant litter[23,24]. Furthermore, plant species litter attracts specific fungal decomposers[23,24], which could explain

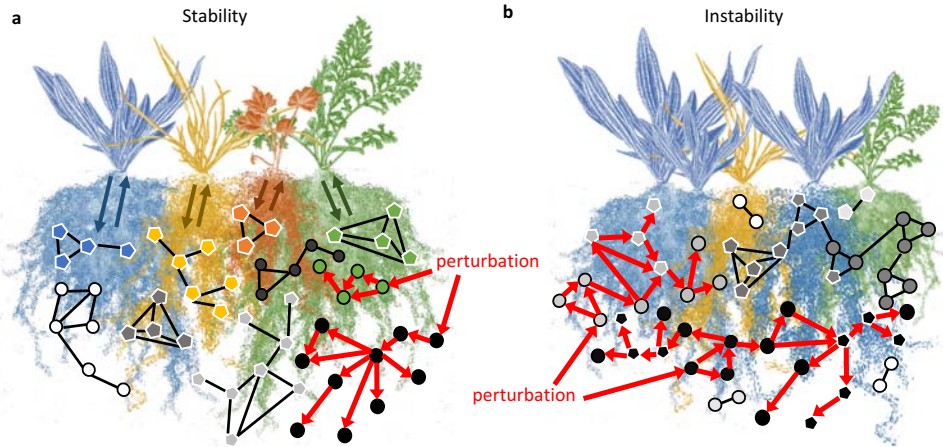

**Fig. 7 | Conceptual framework reconciling the decoupling of prokaryote and fungal responses in buffering the propagation of local perturbation effects. a** In stable settings, prokaryotes (circles, black border) and fungi (pentagons, white border) occur in separate clusters. This separation results from different factors shaping prokaryote and fungal niches. For example, fungal clusters are for an important part directly associated with different plant species (colours of fungal pentagons match colour of the plant species). The resulting microbial networks are likely to buffer local disturbance effects as effects on a subset of the occurring clusters are not likely to spread to unconnected clusters (red arrows showing the spread among connected taxa). **b** In unstable settings, prokaryotes and fungi respond in tandem and create three dominant clusters. This coupling results from a few dominant factors shaping both prokaryote and fungal soil network clusters. The resulting microbial networks are likely to spread disturbance effects throughout large parts of the network given the high connectiveness of dominant clusters (red arrows).

why fungal networks were particularly linked to past compositional pathways in which plant identity plays a key role. In addition, studies demonstrating a successional transformation of fungal communities from early to late decomposers suggests that changes in litter input may result in long-lasting effects[23,24]. We conclude that litter input and decomposition processes likely create long-lasting soil legacies critical for the decoupling of prokaryote and fungal networks, which ensure network stability (Fig. 7).

We further found that putative plant pathogens were consistently associated with past plant community pathways in natural grassland soil. Soil-borne pathogens are increasingly found to be key to plant stabilising processes by preventing dominant plant species from outcompeting less successful plants[2–4]. In line, we found specific network clusters of putative pathogens to be increased when early plant species maintained abundant after the start of plant invasion in 2012. Studies support the idea that negative soil legacy effects increase in strength with plant species time since establishment in a new range[25,26] and similar temporal processes have been suggested to occur within plant communities itself[27]. In addition, we found generalist pathogens to be increased when plant species with a high soil resource optima became more abundant with the start of invasion. This relation aligns with the well-established trade-off between plant species growth and defence, where species with high soil resource needs prioritise growth over defence[28]. Importantly, the specialisation of putative plant pathogens towards various past pathways was absent in abandoned arable soil. Instead, almost all putative plant pathogens were found in a single cluster and associated with a single past pathway. We therefore conclude that plant-mediated soil legacy effects via soil-born pathogen accumulation is likely to play an essential role in ensuring plant community stability.

## Soil chemistry is a critical mediator in coupling microbial networks

We show that soil chemistry is a key mediator in plant-microbiota interactions. Interestingly, the position of soil chemistry in mediating plant community pathways differed between abandoned arable and natural grassland soil. In abandoned arable soil, soil chemical changes largely associated with plant community pathways in the year of sampling, while in natural grassland soil, past plant community pathways largely mediated soil chemical changes. This difference likely resulted from the varying extent of soil resource

depletion in the two soils. Natural grassland soil was more depleted in N and organic C than abandoned arable soil, both at the start of the experiment and after 13 growing seasons[19]. This N depletion leads to a shift towards using more recalcitrant C by soil microbes, causing soil C depletion and a substantial increase in microbial network complexity[29]. Consequently, plants in the year of sampling could have little impact on overall soil chemistry and past litter inputs created strong legacies in natural grassland soil. This is in line with the findings that plant composition in the year of sampling associated with microbial communities either through belowground productivity or direct pathways, which both likely resulted from interactions with plant rhizodeposits such as root exudates[30]. We suggest that soil resource depletion plays a major role in decoupling fungal from prokaryote responses by enhancing soil legacy effects.

Contrary to natural grassland soil, plant composition in the year of sampling had a significant impact on soil chemistry in abandoned arable soil. These changes in soil chemistry associated with both prokaryote and fungal networks, likely coupling their responses. We found that soil N cycling was increased with a high cover of legumes and a high cover of high resource optima species (mainly invaded plant species). Legumes are well-known to enrich N levels in soil through N-fixation, but increased soil N is also commonly observed when ruderal plant species invade communities[21,31]. In the latter case, observed associations with ammonia oxidising archaea implies a crucial role for plant-controlled processes such as rhizosphere priming—the acceleration of organic matter turnover via root exudation—in enhancing soil N cycling and coupling prokaryote and fungal networks[32,33]. Notably, these same plant compositional pathways were correlated with a reduction in fungal specialisation and biomass, implying direct detrimental effects on fungal community complexity. We conclude that soil chemistry and the ability of plants to influence soil resource cycling are crucial mediators in the coupling/decoupling of microbial networks.

## Plant diversity decouples soil microbial networks in natural grassland soil

We show that plant diversity in the year of sampling was a key factor in shaping prokaryotes and fungal communities in abandoned arable soil. In addition to reducing bacterial, fungal and AMF biomass, plant diversity was associated with the dominant prokaryote and fungal

network clusters, thus likely coupling their responses. In contrast, in natural grassland soil, plant diversity increased fungal diversity and directly associated with fungal clusters only. Given that we took six soil cores from each mesocosm and pooled these, it is likely that plant diversity pathways related to the extent of spatial heterogeneity of the communities. Diverse plant communities are inherently more spatially heterogenous above- and belowground, and this is likely to result in higher fungal diversity, as different plant species have unique associated fungi[24,34]. In other words, a higher fungal diversity would be observed when sampling at multiple spaces in a diverse plant community compared to a low diverse community. Importantly, spatial soil heterogeneity has been suggested to be critical in ensuring diversity and stability in plant communities[2,27,35]. We suggest that plant diversity is critical to stability by creating spatial soil heterogeneity and decoupling prokaryote and fungal networks (Fig. 7).

Our findings in abandoned arable soil suggest that plant diversity did not lead to the same level of spatial heterogeneity as observed in natural grassland soil. To understand this, it is important to realise that plant diversity differences in our communities resulted from local plant species loss and invasion processes over time. As these processes are non-random, a low plant diversity likely represented a situation in which certain successful plant strategies dominated, resulting in low plant functional diversity and low soil multifunctionality[36,37]. Hence, plant species in low diverse assemblies on abandoned arable soil likely affected microbial communities in relative similar ways, resulting in a coupling of prokaryote and fungal responses, and instable networks (Fig. 7).

## Conclusion

We found remarkably different topologies in soil microbial networks in plant communities established on natural grassland soil and plant communities established on soil abandoned from extensive agricultural practices 60 years before the start of the experiment. Microbial networks in stable natural grassland soil were largely decoupled in prokaryote and fungal responses, while prokaryote and fungal networks in instable abandoned arable soil largely responded in tandem. We suggest that interactions of microbiota with plants and soil resource cycling underlie the decoupling of prokaryote and fungal networks. Plant community and soil chemical parameters therefore likely provide important, easy to measure predictors of belowground microbial network stability of grasslands globally[5]. Similarly, plant community and soil chemical factors are promising aspects to consider in designing plant communities with decoupling effects on soil microbial networks to increase stability of agricultural systems[38,39]. Future challenges lie in connecting plant and microbial community stability to its driving forces across a multitude of ecosystems, soil resource conditions, land management and perturbation types[5,40].

## Methods

### Experimental design

Experimental plant communities consisting of 44 perennial dry grassland species were sown in May 2007 (Supplementary Table 9)[19]. Plant species were sown in equal proportions with three sowing densities on two soil origins: a dry natural grassland soil (excavated near Encovany, Czech Republic; 50°31′44.6″N, 14°15′12.6″E) and a soil on which dry natural grassland was turned into agricultural land, extensively managed and abandoned 60 years before soil was collected (excavated near Institute of Botany, Czech Academy of Sciences; 50°0′7.11″N, 14°33′20.66″E) ($n = 10$, 60 plant communities in total; Supplementary Fig. 13a). The three sowing densities represent 25%, 100% and 400% of the seed density per m$^{-2}$ as estimated at the natural dry grassland location[19]. Despite significant effects of sowing density in the first three years of the experiment[19], sowing density did not significantly affect above- and belowground plant, microbial and

chemical properties in the current study (data not shown). Sowing density was therefore incorporated as a random effect rather than a fixed factor in all analyses.

Plant communities were grown in 90 L mesocosms (diameter 65 cm, height 36 cm) in the experimental garden of the Institute of Botany, Czech Academy of Sciences[19]. This location provided similar environmental conditions to the natural grassland location. Plant communities did not receive any watering or fertiliser. Only in rare periods of extreme drought when plants showed signs of wilting, communities were watered with rain water.

Importantly, mesocosms were regularly inspected and all species from outside the sown species pool were weeded until September 2011. After this, plant species from the experimental surroundings of the mesocosms were allowed to invade into the established plant communities (Supplementary Fig. 13b; Supplementary Table 9).

### Seed collection and sowing

Seeds of most of the plant species were collected from the natural grassland field site and a further 5% were obtained from nearby seed production fields (Planta Naturalis, Markvartice, Czech Republic). Seeds of all plant species were sown simultaneously. For this, seeds were placed on the soil surface and gently pressed into the soil to avoid seeds from blowing away[19].

### Plant species aboveground proportions

Plant community aboveground biomass was harvested every July and September from 2007 until 2011. From 2012 onward, aboveground biomass was harvested only once a year in July. These time points are similar to management practises at the natural grassland site. Aboveground biomass was cut off 3 cm above the soil and from 2007 until 2011, biomass was sorted per plant species, dried at 60 °C for at least 48 h, after which dry weight was determined. From 2012 onward, plant species biomass was estimated by determining the percentile abundance of each plant species per mesocosm and multiplying this by the total aboveground biomass cut at 3 cm height and dried at 60 °C for at least 48 h. In 2014 and 2015, aboveground biomass was cut, but no species proportions were determined.

### Soil sampling

After the growing season in December 2019, soil cores of 6 cm in diameter and 36 cm length were taken at six random positions in each plant community (Supplementary Fig. 13b). Aboveground plant parts were removed and soil of the six cores was thoroughly mixed by passing it through a 2 mm mesh. Roots that did not pass the mesh were collected, dried at 60 °C for at least 48 h after which dry weight was determined. Subsamples from the mixed soils were taken for soil chemical determination of total nitrogen (N), total and organic C, plant available $NO_3^-$, $NH_4^+$ and $NO_2^-$, K, P and pH. Furthermore, subsamples for analyses of total bacterial, fungal and arbuscular mycorrhizal fungi (AMF) biomass using PLFA and NLFA analyses were taken as well as for microbiome community composition analyses using 16S and ITS amplicon sequencing.

### Soil chemical analyses

All soil was sieved on a 2 mm mesh an thoroughly mixed. Plant available nitrogen (N) (mg kg$^{-1}$ dry soil) was determined by adding 50 mL of 0.5 M $K_2SO_4$ to 5 g of fresh soil, shaking for 30 min and filtering the soil out. $NO_3^-$, $NH_4^+$ and $NO_2^-$ concentrations in the filtrate were determined by Flow Injection Analysis (QuickChem 8000 FIA; Lachat Instruments, Loveland, CO, USA). Plant available P was determined following Olsen et al.[41]. In brief, 5 g air dried soil was extracted with 50 mL of 1 M $NaHCO_3$ adjusted to pH 8.5 with addition of activated carbon to eliminate discoloration resulting from humic acid release. The solution was shaken for 2 h and soil was filtered out. Available P in the filtrate was determined by the Olsen photometric method (ATI

Unicam UV 400/VIS Spectrophotometer at 630 nm)[41]. K was determined in 5 g air dried soil by shaking with 50 mL Mehlich II soil extraction solution (Hach Lange GmbH, Düsseldorf, Germany) for 30 min. Soil was filtered out and and Mg, Ca, and K were measured in the filtrate using atomic absorption spectrometry (ContrAA 700 with $C_2H_2$-air flame for Mg and K, and $C_2H_2$-$N_2O$ for Ca; Analytik Jena GmbH, Jena, Germany). Exchangeable pH was measured in a solution of 5 mL in 25 mL 0.1 M KCl shaken for 30 min (WTW Multilab 540; Xylem Analytics, Weilheim, Germany). Total N, C and organic C were determined in dried soil ground to <0.1 mm particle size using combustion analyses (FLASH 2000 CHNS/O Analyzer; Thermo Fisher Scientific, Waltham, MA, USA).

## Soil bacterial and fungal biomass

Soil bacterial and fungal biomass was determined using PLFA and NLFA analysis following García-Sánchez et al.[42]. In short, 1 g of fresh soil taken from the mixed soil cores was freeze-dried in a chloroform-methanol-phosphate buffer (1:2:0.8, v/v/v)[43]. Lipids were fractioned into polar lipids (PLFAs), glycolipids and neutral lipids (NLFAs), using an extraction cartridge (LiChrolut Si-60; Merck KGaA, Darmstadt, Germany) and subjected to alkaline methanolysis[44]. Following Sampedro et al.[45], free methyl esters of PFLAs and NLFAs were analysed by gas chromatography-mass spectrometry (450-GC with 240-MS IT Mass Spectrometer; Varian Medical Systems Inc., Palo Alto, CA, USA). Total microbial biomass was estimated by the sum of all PLFA contents. Bacterial biomass was based on the summed PLFA contents i14:0, i15:0, a15:0, 16:1ω5, 16:1ω7; 16:1ω9, 10Me-16:0, i16:0, i17:0, a17:0, cy17:0, 17:0, 10Me-17:0, 18:1ω7, 10Me-18:0 and cy19:0, and actinobacterial biomass based on the summed contents 10Me-16:0, 10Me-17:0 and 10-Me18:0. Gram-positive and gram-negative bacterial were quantified based on i14:0, i15:0, a15:0, i16:0, i17:0, a17:0 and 16:1ω7, 16:1ω9, 18:1ω7, cy17:0, cy19:0, respectively. Fungal biomass was quantified based on PLFA content 18:2ω6,9[44] and NLFA 16:1ω5 was used as a marker for AM fungi[46].

## 16S and ITS amplicon sequencing

All frozen soil samples (250 mg each, in duplicates for each sample) were homogenized and lysed in PowerBead Pro Tubes (Qiagen, Germany) on a Vortex adapter. Subsequently, DNA was extracted using the DNeasy PowerSoil Kit (Qiagen, Germany) according to the manufacturer's instructions and eluted in 50 µl of elution buffer. The fungal internal transcribed spacer of the rDNA (ITS2 rDNA) was amplified using primers gITS7ngs[47] and ITS4[48]. The bacterial 16S rRNA gene (V4 region) was amplified from the same DNA extracts using primers 515f and 806r[49] (see Supplementary Data 2 for primer sequences). All primers were tagged with sample-specific barcodes of 10–12 bases. PCR mix was performed in the total volume of 15 µl and contained 0.07 U Thermo Scientific™ *Taq* DNA Polymerase, 10x PCR Buffer, 2.5 mM $MgCl_2$, 20 µg BSA (all Thermo Fisher Scientific, Waltham, Massachusetts, USA), 0.3 mM each dNTP, 0.3 µM of each primer and 1 µl of the DNA extract. Thermocycling conditions were 94 °C for 4 min, 25 cycles of 94 °C for 45 s, 52 °C for 60 s and 72 °C for 35 s, followed by 10 min at 72 °C. Each DNA extract was amplified in duplicate. PCR products were visualized on a 1% agarose gel. The pooled duplicates were purified through columns with the QIAquick PCR Purification Kit (Qiagen, Hilden, Germany) according to the manufacturer's protocol and eluted into 20 µl of ddH₂O. DNA concentrations of the amplicon pools were quantified using a Qubit 2.0 Fluorometer (Thermo Fisher Scientific) with High Sensitivity Assay Kit. The purified amplicons were pooled in equimolar ratios. Both negative PCR controls (with ddH₂O instead of a template) were processed in the same way as the experimental samples and included into the sequencing library, together with sixty fungal and sixty bacterial amplicons. The library was sequenced on an Illumina MiSeq instrument (2 × 250 bp) (SEQme, Dobříš, Czech Republic).

## Plant community diversity and composition

All analyses were performed in R version 3.6.1[50]. Plant community diversity was calculated based on the Shannon index using the function diversity of the vegan package[51]. Plant community temporal stability was calculated based on the inverse coefficient of variation of total aboveground productivity over time using the package codyn[52]. The first year of data was removed for this calculation as the communities were still establishing, which resulted in a disproportionally large fluctuation in aboveground productivity (Fig. 1b).

Plant community compositional changes were assessed using detrended correspondence analysis (DCA). For this, aboveground biomass of species with in total >10 observations over all replicates and all years was analysed using decorana from the vegan package[51]. To understand to which general processes the compositional changes associated, we created a passive overlay of plant species ecological optima (Czech Ellenberg indicator values for soil nitrogen, moisture and reaction[53]) and plant species average residence period (average year of presence between 2012 and 2019) for DCA scores 1 and 2 using envfit of the vegan package[51] (Supplementary Fig. 1a). Permutations (999 permutations) were used to test for significance. DCA score 3 was not significantly related to any plant traits (data not shown), but to a change in legume cover (Supplementary Fig. 1b).

## Past plant community parameters

To describe variation in plant community development over time, we calculated two types of past plant community parameters: the initial impact of the start of plant species invasion on the plant community and developmental trajectories after the start of invasion. These two types of past parameters were calculated for plant community aboveground productivity, plant diversity (Shannon index) and the DCA scores 1–3. For the initial impact of plant invasion, we calculated the difference between parameters values in 2011 (last year without invasion) and 2012 (first year with invasion). This was with exception of plant diversity for which drastic changes due to the start of invasion occurred over a period of two years and the one-year difference did not relate significantly to any soil or microbial measurements in 2019 (data not shown). Initial invasion impact for plant diversity was therefore calculated between 2011 and 2013. Developmental trajectories were calculated as overall increase/decrease trends of each plant community parameter from 2012 until 2019. For this, we fitted a linear model using lm and extracted the beta slope using lm.beta[50].

## 16S and ITS bioinformatics

In total, Illumina paired end sequencing of 120 samples (60 16S and 60 ITS amplicons) and two negative controls yielded 4,261,236 raw sequences. The data were processed using the pipeline SEED2 ver. 2.1.1b[54]. First, low-quality sequences were discarded (mean quality score <30). The reads were demultiplexed (no mismatch allowed in the tag sequences) and tag switches (i.e. reads with non-matching tags) were discarded.

The ITS2 region was extracted from the fungal amplicons using ITSx ver. 1.0.11[55] and sequences shorter than 20 bp were excluded. This yielded 982 036 sequences which were clustered to OTUs using UPARSE implementation in USEARCH ver. 8.1.1861[56] with 97% similarity threshold (45 480 chimeric sequences were excluded during this step). The most abundant sequences were selected for each of the resulting 10,685 OTUs. These sequences were checked for their identity via BLASTn algorithm against the UNITE database ver. 8.3[57] and non-fungal, no-hit sequences as well as global singletons, doubletons and tripletons were excluded from further analyses leaving 2638 OTUs represented by 840,206 reads. Six reads detected in the negative control were subtracted from the corresponding two OTUs across the dataset. The ecological guilds of the fungal OTUs were parsed using the database FungalTraits[58] at genus and sequence levels. All abundant OTUs were assigned with high probability.

Primers were cut from prokaryote reads (1,319,594 reads after demultiplexing) and sequences shorter than 20 bp were excluded. The reads were clustered to OTUs using UPARSE implementation in USEARCH ver. 8.1.1861[56] with 97% similarity threshold (442,826 chimeric sequences were excluded during this step). OTUs with $n < 5$ were discarded. The most abundant sequences were selected for each of the resulting 6532 OTUs. These sequences were checked for their identity via BLASTn algorithm against the RDP trainset 16[59]. 179 reads detected in the negative control were subtracted from the corresponding OTUs across the dataset. OTUs with non-target and no BLASTn hits were excluded from further analyses leaving 6369 OTUs represented by 841,512 reads.

## Microbial community analyses

Significant separation between soil origins was tested with PERMANOVA using adonis from the vegan package[51] based on centered log ratio (clr) transformed read counts and visualised using PCA with the phyloseq package[60]. Alpha-diversity (Shannon diversity) was calculated based on multiple rarefaction (1000 iterations) using the metagMisc package[61]. The overlap in OTUs between natural grassland and abandoned arable soil for prokaryotes and fungi was calculated using ps_venn[62].

For each OTU, we calculated a specialisation index (SI)[63]. SI of the $i$th OTU was calculated as the coefficient of variation minus a correction for under-sampling of rare OTUs:

$$SI_i = \left(\frac{\sigma_i}{\mu_i}\right) - \sqrt{\frac{K}{N_i}} \qquad (1)$$

with $\sigma_i$ being the standard deviation of the reads of the $i$th OTU across all samples, $\mu_i$ being the mean of the reads of the $i$th OTU across all samples, $K$ the number of habitat classes (2 soils) and $N$ the total number of reads of the $i$th OTU across all samples. Calculations were based on rarefied read abundances (rarefied to the smallest sample size: 4517 and 1618 for 16S and ITS, respectively). Average SI of each sample was calculated as the community weighted mean of the SI of all OTUs present in the sample.

## Microbial network analysis

We constructed co-occurrence networks across the 30 plant communities on each soil origin for both prokaryote (bacteria and archaea) and fungal communities. We first filtered each dataset to exclude rare OTUs with total <100 reads and OTUs that were present in <5 samples per soil origin. Co-occurrence networks were then calculated using the SpiecEasi package[64]. In brief, networks were inferred based on clr transformed read counts, neighbourhood selection (MB method) and we selected optimal stability parameters using the StARS selection approach (threshold 0.05, nlambda 20 with 999 replications)[65]. We clustered similarly responding OTUs in each network using the Spinglass algorithm of the igraph package[66–69]. This approach clusters OTUs based on both positive and negative edges as well as their weight. Present and absent edges as well as positive and negative edges were given a similar importance, and unlimited spins (clusters) were provided. Relative read counts were summed per cluster per sample and used in further correlation and structural equation models (SEM).

We tested whether networks were (i) significantly more or less densely clustered, and (ii) significantly more or less coupled in prokaryote and fungal clusters than based on chance using a null-model approach. For this, we rewired the original networks edges with preserving the original networks degree distribution using the robin package[70]. We (i) calculated the clustering coefficient (modularity), and (ii) the number of significant correlations occurring between prokaryote and fungal clusters of 1000 rewired networks. $P$-values were calculated as the proportion of randomised (i) clustering coefficients and (ii) number of significant correlations smaller than in the original networks[3,71].

The coupling/decoupling of prokaryotes and fungi within each plant community was described by the beta slope of a generalised linear model (glm) between the summed relative read counts in the most prevalent prokaryote 1–9 and fungal clusters 1–9. This correlation is valid as cluster assignment within the microbial networks is based on the order of prevalence of the ASVs in each cluster, hence the similarity in patterns of cluster sizes that occurred across different networks (Fig. 3a, b). Summed relative read counts per cluster were ln- or sqrt-transformed when model residuals did not follow a normal distribution. The beta slope was extracted using lm.beta[50]. A positive value indicates that prokaryote and fungal reads were distributed similarly over the most prevalent network clusters, and thus, that prokaryote and fungal responses were coupled. A value of zero indicates that prokaryote and fungal read distribution over the network clusters was unrelated and thus decoupled. A few weak, negative correlations occurred and indicate that prokaryote and fungal read distribution over the network clusters were nearing an opposite pattern. These indicated cases of a strong decoupling of prokaryote and fungal responses.

## Microbial network cluster characteristics

We inferred putative dominant metabolic and functional characteristics of each microbial network cluster. This was done based on similarities, if present, in knowledge of the prokaryote or fungal taxa co-occurring in each network cluster: putative functions were obtained from literature and the FungalTraits database[58], relative habitat specialisation, and enrichment in natural grassland or abandoned arable soil (Supplementary Figs. 9–10; Supplementary Tables 2–5). Relative habitat specialisation of network clusters was calculated by taking the community weighted mean of the SI of each sample in each cluster (see Microbial community analysis). Relative habitat generalist clusters were defined as clusters below the community-wide mean SI without overlap of the maximum distribution (75th percentile + 1.5 * interquartile range). Relative habitat specialists were defined as clusters above the community-wide mean SI without overlap of the minimum distribution (25th percentile − 1.5 * interquartile range) (Supplementary Fig. 9)[63].

Cluster enrichment in either of the soils was calculated by taking the ln-response ratio of read counts (rarefied) in natural grassland soil divided by read counts in abandoned arable soil for each OTU. Network clusters with response ratios above zero and without overlap of the minimum distribution were defined as enriched in natural grassland soil. Network clusters with response ratios below zero and without overlap of the maximum distribution were defined as enriched in abandoned arable soil (Supplementary Fig. 10).

## Overall statistics

Analysis of variances were performed using linear mixed-effects models in R version 3.6.1[50]. All models in which time (continuous, scaled) and soil origin were included as an explanatory variable, included mesocosm and sowing density as random effects to take repeated measures and initial sowing densities into account. Models were fitted using lmer of the lme4 package[72]. Mesocosm position in the garden had negligible effects, in many cases resulted in overfitting of the models and was therefore dropped as a random effect.

Correlations between plant, soil chemistry and microbial parameters including network clusters were performed using lme of the nlme package[73] and included sowing density as a random factor. For these multiple correlations, $p$-values were corrected for multiple testing using the Bonferroni correction[74]. In all models, data was ln- or sqrt-transformed when model residuals did not follow a normal distribution. In case of heterogeneity of variances, data weighting per soil origin using varIdent from the nlme package was incorporated[75].

## Structural equation modelling

We hypothesised that plant community parameters in the year of sampling and developmental trajectories in time affected microbial communities with soil chemical changes as a potentially important mediator. We used structural equation models (SEM) which allowed us to separate direct pathways between plant communities and soil microbiota from indirect pathways via soil chemical changes (Fig. 5a). We note that the directionality of soil chemistry onto microbial communities is a mathematical necessity to test whether soil chemistry was a significant mediator. Such pathways should however be interpreted as an interaction as both soil microbiota and plant communities influence soil chemistry.

All SEMs were fit using piecewiseSEM[76] and lme of the nlme package[73] with initial sowing density as a random effect ($n = 30$). Overall model fit was assessed using direction separation tests (d-sep) based on Fisher's C statistics with models being accepted if $p > 0.1$. We simplified our models using a backward stepwise elimination procedure for which we consecutively removed pathways with the highest $p$-value[77]. Endogenous variables were allowed to drop from the model in case effects were not significant ($p > 0.05$). The model with the lowest Akaike information criterion (corrected for small sample sizes; AICc) was then selected as the best fit base model.

To keep the number of potential pathways relative low and avoid spurious effects occurring due to correlating exogeneous variables, we first calculated three base models for each soil origin[78]. These three base models captured effects of the plant community onto soil chemical changes after the 13th growing season of (a) the plant community in the year of sampling (aboveground productivity, plant diversity and plant compositional DCA axes 1−3), (b) overall effects of the plant community from the past (initial invasion effect on plant diversity, aboveground productivity and plant diversity trajectories in time), and (c) plant compositional effects from the past (invasion effect size on plant compositional DCA axes 1−3, plant compositional DCA axes 1−3 trajectories in time). All base models included the same soil chemical parameters: total soil N, organic C and pH, and plant available P, $NO_3^-$, $NH_4^+$, $NO_2^-$ and belowground productivity. The three base models were then combined. Each microbial network cluster (summed relative reads per 16S or ITS cluster from the calculated co-occurrence networks per soil origin), bacterial, fungal and microbial biomass (PLFA/NLFA analyses) as well as alpha-diversity indices and average SI of the microbial communities were ran through the SEM model as the final parameter to be estimated. Per run, one microbial parameter was considered, which could be affected either directly by the plant community parameters or indirectly via soil chemical changes. Each run, a backward stepwise elimination procedure to consecutively remove non-significant pathways was followed in the same way as performed for the base models[77]. All microbial variables not following a normal distribution were ln- or sqrt-transformed.

## Relative contribution calculation

We created 64 unique SEM models (3 microbial biomass pools natural grassland soil + 3 microbial biomass pools abandoned arable soil + 9 16S clusters natural grassland soil + 10 16S clusters abandoned arable soil + 21 ITS clusters natural grassland soil + 18 ITS clusters abandoned arable soil). We extracted the direction and effect size of each significant pathway of each model ($p < 0.05$) and calculated the contribution of each plant parameter to changes in microbial biomass and microbial co-occurrence clusters. For each significant, direct plant-microbial pathway, we multiplied the path effect size with the relative size of the microbial pool or cluster it was affecting. This multiplication sized the pathway effect to its relative importance to the microbial community as a whole. For indirect pathways, we multiplied the effect size of the plant parameter onto the soil chemical variable with the effect size of the soil chemical variable onto the microbial variable. The obtained effect sizes were

scaled to the relative size of the microbial variable they were affecting using community weighted means.

To create an overview of all effects, the relative contribution of each plant parameter per microbial variable was summed per group: all combinations of plant parameters in the year of sampling versus past parameters, overall versus compositional parameters, and direct versus indirect pathways (Fig. 5a). These grouped effect sizes were scaled to the number of potential pathways within each group to allow a direct comparison of the relative contribution of each plant parameter group (Fig. 5b).

### Reporting summary

Further information on research design is available in the Nature Portfolio Reporting Summary linked to this article.

## Data availability

All raw data generated in this study have been deposited in the Zenodo digital data repository (https://doi.org/10.5281/zenodo.6695065)[79]. The raw microbial sequencing data generated in this study have been deposited in the NCBI SRA database under BioProject ID PRJNA931221. Fungal trait data is publicly available via the FungalTrait database[58]. Czech Ellenberg values for flora are publicly available via Chytrý et al. [53]. Source data are provided with this paper.

## Code availability

All R scripts are publicly available via Github (https://doi.org/10.5281/zenodo.8032393)[80].

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

## Acknowledgements

We are grateful to Andrea Jarošová and Mária Šurinová for preparing the
samples for amplicon sequencing. We thank Michiel in 't Zandt for his
knowledge on prokaryote families and Francisco Dini-Andreote for
answering questions on microbial co-occurrence network analysis. D.Z.
is grateful to Michael Bahn and Fiona Brennan for their support and
hosting her at their groups. D.Z. was supported by a PPLZ Postdoctoral
Fellowship awarded by the Czech Academy of Sciences. Z.M.
acknowledges funding from The Czech Science Foundation (GAČR 20-
01813S), long-term research development project RVO 67985939 of the
Czech Academy of Sciences and the Ministry of Education of the Czech
Republic (MŠMT).

## Author contributions

Z.M. designed and set up the experiment; Z.M. maintained the experi-
ment and led the sampling campaigns; T.C. analysed soil samples for
PLFA/NLFAs; Z.K. performed the downstream microbial bioinformatics;
D.Z. performed all further microbial, soil and plant data analyses and
statistics; D.Z. wrote the publication, all others co-commented. All
authors contributed critically to the manuscript and gave final approval
for publication.

## Competing interests

The authors declare no competing interests.

## Additional information

**Supplementary information** The online version contains
supplementary material available at

Dina in 't Zandt.

**Peer review information** *Nature Communications* thanks Nico Eisen-
hauer, Emilia Hannula and Raul Ochoa-Hueso for their contribution to
the peer review of this work. A peer review file is available.

**Publisher's note** Springer Nature remains neutral with regard to
jurisdictional claims in published maps and institutional affiliations.

