## [Peer Review File · Nature Communications]

Reviewers' Comments:

Reviewer #1:

Remarks to the Author:

The present study explored associations between plant and soil microbial communities in two different soils in field mesocosms. Plant communities established over a period of 13 years, allowing to test current and plant community effects on soil properties and microbial communities. Soil microbial network properties and their drivers were assessed. Abandoned arable soil, in contrast to natural grassland soil, showed more pronounced variations in plant and microbial community properties, which was interpreted to reflect instability. The authors conclude that plant community stability is associated with soil microbial networks with a high niche differentiation. In general, the manuscript is very well written and tackles an important research question. However, I am wondering about the conceptual advances due to the descriptive nature of the study. (1) How much is the degree of coupling versus decoupling related to the general levels of variation in plant community properties? Figure S1 suggests to me that there might have been higher stochasticity in plant community development and a higher dissimilarity of communities in abandoned soil in comparison to grassland soil, probably spanning larger gradients in biotic and abiotic conditions, which might affect the coupling results. (2) A lot of the presentation and discussion is about direct versus indirect plant community effects on soil microbes. However, it remains fairly open what these direct effects could be. For me, "direct effects" in SEM may mostly indicate that critical variables mediating effects were not assessed. This is totally fine, as long as potential mechanisms are discussed and hypotheses presented. I would speculate, for instance, that these direct effects may be mediated by rhizodeposits. (3) While some of the wording is cautious given the unknown directionality of effects, some presentation is surprisingly "directional", such as the case in Figure 5. For example, soil chemical properties are assumed to influence soil microbial communities, while it is well known that the opposite might also be true and reciprocal effects occur. (4) Figure 6 introduces a hypothesis how microbial network properties and community compositions may be associated with stability during perturbations. I am wondering if this concept is sufficient given that microbial community traits have not been included. For example, microbial community resistance to perturbations may not only be determined by network structure, but also by the dominance of different microbial taxa like soil fungi. (5) Conclusions: I am also wondering if the predictability of soil microbial properties based on plant community composition in this mesocosm study can really provide general principles, given that strong plant identity effects may co-determine soil microbial properties and feedback effects. (6) Lines 57-75: strong focus on identity/sampling effects; how about the potential effects of complementarity and facilitation?

Reviewer #2:

Remarks to the Author:

The paper by Dina in 't Zandt et al. reports on the results of a very interesting mesocosm experiment carried out over the course of 13 years in which plant communities were developed from soils with two origins: grassland and ex-arable field. The results reported are highly intriguing and provide a novel perspective on the link between plant community stability and the coupling of soil microbial communities (fungi and prokaryotes).

Despite the potential of the article to be a nice contribution to the field, I also found the article dense and hard to read at times due to the prolific description of many results on various aspects linked to the structure of plant and microbial communities that resulted in the presentation of an uncoherent story at times. Or, in other words, the main goal and message of the paper were decoupled from the overall structure of the paper, perhaps due to the high amount of data available and the complexity of the experimental design. Also, the interpretation of the decoupling of fungal and prokaryotic communities is not satisfactory to me. The authors claim a very important role of niche partitioning, yet this niche partitioning would imply competition/cooperation among microbial taxa as the main driver of the self-organized patterns of microbial communities. Yet, the authors simultaneously argue that plant communities and particularly their stability is the main driver of microbial network coupling/decoupling. It is important in my opinion to reconcile these two aspects.

L22-23. Interesting, but unsure what it exactly means as written?

L24. Why is this so?

L30 More like associate with.

L40-56. This is an unclear paragraph mixing many concepts but not landing anywhere.

L41. Perhaps introduce the concept of plant soil feedbacks

L42-43. Unfinished sentence

L43-45. Complex sentence. Can you say this is a more direct and concise way?

L57-75. Again, this is a convoluted paragraph that attempts to link the stability of plant and microbial communities, but it does so in an ineffective way.

L57. Citations for this are needed.

L135-136. But how were these values at the beginning of the study? Were they also different and remained different?

L149-150. How did you determine what OTUs belonged to each cluster?

L162-164. Very interesting result. Can you tell whether microbial communities are more, equally or less coupled than by chance? You can do this by using a null modelling approach. The framework in Ochoa-Hueso (2021), *One Earth*, may be useful for that.

L180. I found the results associated with this paragraph very complex to understand, in part because Fig 4 is not particularly informative/clear.

L186. What do you mean?

L188-189. The meaning of this is unclear.

Figure 4. The meaning of this figure is unclear to me.

L263. Was not it the other way around? Now I am confused.

L282-283. This is not necessarily linked with niche occupancy. It simply means that fungal and prokaryotic communities are less spatially structured in grassland soils, which may have to do with the destabilizing role of plant communities developed over these soils as compared to those growing on exarable land soils.

Reviewer #3:

Remarks to the Author:

This is a manuscript on microbial networks in abandoned and natural grasslands. Authors have made a clever experiment which allows them to link plant community stability and microbiomes. The manuscript is well written and complicated analyses presented in a simple way. Especially figure 6 is wonderful. The topic investigated is pretty novel and here many new aspects are presented. This will contribute to the plant literature although drawing causality from this data is not advised.

I really like this manuscript, but do think certain aspects could be investigated with more statistical analysis and rigor. Especially the microbial side is pretty light while plant theory is well established. It is further a pity that microbes are analysed only once and that rhizosphere was not included in the analysis. Rhizosphere is the part of soil that would be in theory affected more by current plants while bulk soil more by previous plants (and does contain more dead microbes still having DNA). This is something that cannot be addressed here due to lack of sampling points and compartments.

Statistical issues:

NMDS is not a suitable method to investigate differences between loadings (i.e. residence period, residence time). Here also more statistically robust methods should be used to evaluate the differences explained in community structure. This is pretty descriptive now. It would be important to see the statistics for the full NMDS and not per axis, for visual purposes it is ok to use them but this needs further interpretation with proper statistical tests. Correlating NMDS (that consist that parameter) with a parameter is not a neat way to analyse data.

Using networks and clusters shows more or less the same results as would be captured with more traditional analysis. In general the ms is quite descriptive and light on statistics. However, the figures are clear.

Interestingly, for microbes PcoA was used. This would be a suitable method also for plants. For microbes, it would have been good to know how different the communities were at the beginning and not only after 13 cycles. The soil chemistry was described in the beginning but we have no information on soil microbiome at the beginning of the experiment. This could be potentially one of the driving forces explaining some of the variation.

Other points:

The results on past plant communities predicting soil microbial community are interesting. However, microbes were measured only once and they might fluctuate as well yearly. In the SEM link between plants and microbes is made but it is unclear if these effects of especially past plants are direct or indirect (plants affecting other plants or soil abiotics). The effects are nicely shown but some links (plants on plants for example and between fungi and bacteria) are ignored. Figure 5 is intriguing, it is not easy to understand as it is something between a network and a model. It does show clearly how complex the situation is and how all the parameters are connected.

Table 1 is quite dense in information and takes some moments to go through. Here more graphical representation would be advised.

It is interesting that fungi and bacteria would fully occupy separate soil niches. There must be large overlap for certain species while for others much less. If we talk about niches we should also acknowledge the scale we look at. For bacteria a whole soil sample is the size of an universe. We do not actually know where it is located in the soil (in water, in air, on an aggregate, inside an aggregate etc etc). Therefore it is tricky to discuss niche theory here.

Figure 6 is wonderful!

Line 325: it would be helpful to use similar wording throughout. Here 'soil origin' is confusing.

Line 477: the bioinformatic analysis could be explained more clearly. How was Bray-curtis calculated on centerED log ratio? This is quite unusual way to analyse data.

Permanova is done but not reported?

As the putative life strategies of fungi and 'functions' of bacteria are included in the text, it would be very nice to know how authors derived these without digging deep into the supplementary materials. In supplementary materials also it is only said FungalTraits are used and nothing about confidence of the assignment. For bacteria nothing is said about functions yet in the text bacterial functions are discussed and appear suddenly in supplementary tables. Here either add references or use a source for all assignments.

In general, as microbes play an important role in the manuscript, it would be very important to balance also in mat&met between plant and microbiome information.

The definition of the clusters should be explained better as it is not really visible in figure S5. Interestingly for abandoned grasslands almost all clusters only contain species characterized as 'other' (Fig. S6). What could be a reason for this?

Reviewer comments

Reviewer #1 (Remarks to the Author):

The present study explored associations between plant and soil microbial communities in two different soils in field mesocosms. Plant communities established over a period of 13 years, allowing to test current and plant community effects on soil properties and microbial communities. Soil microbial network properties and their drivers were assessed. Abandoned arable soil, in contrast to natural grassland soil, showed more pronounced variations in plant and microbial community properties, which was interpreted to reflect instability. The authors conclude that plant community stability is associated with soil microbial networks with a high niche differentiation. In general, the manuscript is very well written and tackles an important research question. However, I am wondering about the conceptual advances due to the descriptive nature of the study. (1) How much is the degree of coupling versus decoupling related to the general levels of variation in plant community properties? Figure S1 suggests to me that there might have been higher stochasticity in plant community development and a higher dissimilarity of communities in abandoned soil in comparison to grassland soil, probably spanning larger gradients in biotic and abiotic conditions, which might affect the coupling results.

Thank you very much! The reviewer raises an important point here. It could indeed be expected that more variation would occur in the abandoned arable soil communities in plant community, soil chemistry and/or microbial community parameters given that these communities were more strongly affected by plant species invasion. We compared the levels of variation in plant, microbial and soil parameters between the two soil origins to see whether large differences occurred, but the levels of variation between the two soils was surprisingly similar. Moreover, there was also no general trend in the small difference in levels of variation: in some cases the level of variation was somewhat higher in abandoned arable soil, while in other cases the level of variation was somewhat higher in natural grassland soil. We do note that the underlying levels of variation were not clearly visible in our previous plots. We therefore changed all plots to incorporate all individual plant community points per soil origin, so the spread between individual communities can now be seen from the plots (see Fig 1, S2, S8, S9).

(2) A lot of the presentation and discussion is about direct versus indirect plant community effects on soil microbes. However, it remains fairly open what these direct effects could be. For me, “direct effects” in SEM may mostly indicate that critical variables mediating effects were not assessed. This is totally fine, as long as potential mechanisms are discussed and hypotheses presented. I would speculate, for instance, that these direct effects may be mediated by rhizodeposits.

We fully agree. We incorporated a section on the direct effects of past plant community pathways in L297-307 and L312-322. In addition, we included a section on the direct effects of plant community effects in the year of sampling in L335-338, which highlights the likely mediating role of rhizodeposits in these direct plant community pathways.

(3) While some of the wording is cautious given the unknown directionality of effects, some presentation is surprisingly “directional”, such as the case in Figure 5. For example, soil chemical properties are assumed to influence soil microbial communities, while it is well known that the opposite might also be true and reciprocal effects occur.

The reviewer is quite right, both plants and microbiota influence soil chemical composition and we cannot fully distinguish these interactions and feedbacks. We agree that in parts of the manuscript we have not addressed this carefully enough and have reworded these sections throughout the manuscript. In stead of ‘effects’, we now address ‘interactions’, ‘associations’ and ‘pathways’. In addition, we note in the materials and methods that the strict directionality of the pathways in the structural equation models is a mathematical necessity to calculate whether soil chemistry was a

mediator of plant-microbiota interactions, but that these directional effects should be interpreted with caution as both plants and microbiota influence soil chemistry (L560-563). We added a similar note to Figure 4 where we explain the structural equation model approach.

(4) Figure 6 introduces a hypothesis how microbial network properties and community compositions may be associated with stability during perturbations. I am wondering if this concept is sufficient given that microbial community traits have not been included. For example, microbial community resistance to perturbations may not only be determined by network structure, but also by the dominance of different microbial taxa like soil fungi.

This is a valid point. We agree that both network structure and dominance of different microbial taxa are likely to be important in plant community responses to perturbations. To address differences in dominant microbial taxa, we now present more broadly the prokaryote and fungi that were involved in the most important pathways of our structural equation models (L206-259). We rewrote large sections of the discussion to balance the focus on network structure with effects of the microbial groups that were consistently highlighted in the result section: putative soil saprotrophs (L293-307), putative plant pathogens (L308-322), and nitrogen-cycling taxa (L340-352).

In addition, we calculated the overlap between OTUs in the two soils, which showed that hardly any unique OTUs occurred in either soil origin (Fig S3; L129-130) and thus that soil microbial communities significantly separated based on OTU abundance rather than presence/absence. We therefore determined which microbial network clusters were enrichment in natural grassland and abandoned arable soil (Fig S9) and determined whether any consistencies in cluster responses occurred: whether certain pathways consistently associated with natural grassland enriched or abandoned arable enriched clusters (Table S7-S8). This was however not the case.

(5) Conclusions: I am also wondering if the predictability of soil microbial properties based on plant community composition in this mesocosm study can really provide general principles, given that strong plant identity effects may co-determine soil microbial properties and feedback effects.

We agree with the reviewer. This is why we used a structural equation modelling approach to dissect overall plant community effects from plant community compositional effects where plant identity plays a strong role. Importantly, we show generalities in the plant compositional elements. So although plant identity played an important role, we were able to link compositional changes to general concepts: plant community turnover (compositional axes 1), ecological soil resource optima (compositional axes 2) and community legume cover (compositional axes 3). As a result, both overall plant community parameters (diversity and productivity) as well as compositional parameters provide general principles. We do believe this is a critical notion given that plant community composition is often excluded in studies, but plays an essential part as we show here.

(6) Lines 57-75: strong focus on identity/sampling effects; how about the potential effects of complementarity and facilitation?

We agree with the reviewer. Based on comments by reviewer 2, we deleted these few sentence from the introduction as it introduced concepts that were not further explored in the manuscript.

Reviewer #2 (Remarks to the Author):

The paper by Dina in 't Zandt et al. reports on the results of a very interesting mesocosm experiment carried out over the course of 13 years in which plant communities were developed from soils with two origins: grassland and ex-arable field. The results reported are highly intriguing and provide a novel perspective on the link between plant community stability and the coupling of soil microbial communities (fungi and prokaryotes).

Despite the potential of the article to be a nice contribution to the field, I also found the article dense and hard to read at times due to the prolific description of many results on various aspects linked to

the structure of plant and microbial communities that resulted in the presentation of an uncoherent story at times. Or, in other words, the main goal and message of the paper were decoupled from the overall structure of the paper, perhaps due to the high amount of data available and the complexity of the experimental design.

Thank you! We agree that the manuscript had somewhat been invaded by analyses that were decoupled from the original aim. Throughout the introduction, we identified and deleted the concepts that were not further discussed in the manuscript. We further matched the structure of the last paragraph of the introduction (L81-93) with the structure used in the result section. Furthermore, we rewrote large parts of the discussion, which now matches the concepts introduced in the beginning of the manuscript. Please see our more detailed responses to specific paragraph below.

Also, the interpretation of the decoupling of fungal and prokaryotic communities is not satisfactory to me. The authors claim a very important role of niche partitioning, yet this niche partitioning would imply competition/cooperation among microbial taxa as the main driver of the self-organized patterns of microbial communities. Yet, the authors simultaneously argue that plant communities and particularly their stability is the main driver of microbial network coupling/decoupling. It is important in my opinion to reconcile these two aspects.

We understand the reviewers concern and completely deleted the niche partitioning hypothesis from the manuscript. We have rewritten most of the discussion and now focus on plant litter/decomposition effects, putative plant pathogen accumulation, soil resource depletion and spatial heterogeneity as likely mechanistic pathways underlying the coupling/decoupling of prokaryote and fungal responses (see also comments reviewer 1).

L22-23. Interesting, but unsure what it exactly means as written?

We have reworded the sentence, L17-18.

L24. Why is this so?

We clarified the sentence, L21-23.

L30 More like associate with.

Adjusted accordingly (L27).

L40-56. This is an unclear paragraph mixing many concepts but not landing anywhere.

We reduced the number of concepts that are introduced in this paragraph and reworded various sentences to increase its clarity (L37-42).

L41. Perhaps introduce the concept of plant soil feedbacks

The paragraph introduces general network theory and its stabilising attributes. This does not per definition relate to plant-soil feedback, so we do not think this would be a good fit here.

L42-43. Unfinished sentence

The sentence was adapted (L37-38).

L43-45. Complex sentence. Can you say this is a more direct and concise way?

We revised the sentence to be direct and concise (L38-40).

L57-75. Again, this is a convolute paragraph that attempts to link the stability of plant and microbial communities, but it does so in an ineffective way.

We revised the paragraph and removed concepts not further addressed in the manuscript, L52-65.

L57. Citations for this are needed.

The sentence was deleted.

L135-136. But how were these values at the beginning of the study? Were they also different and remained different?

Yes, they were, see L399-400. We now also incorporated this notion in the discussion, L330-332.

L149-150. How did you determine what OTUs belonged to each cluster?

The detailed methods are presented in the Material and methods section, L512-515: "We clustered similarly responding OTUs in each network using the Spin-glass algorithm of the igraph package. This approach clusters OTUs based on both positive and negative edges as well as their weight. Present and absent edges as well as positive and negative edges were given a similar importance, and unlimited spins (clusters) were provided."

L162-164. Very interesting result. Can you tell whether microbial communities are more, equally or less coupled than by chance? You can do this by using a null modelling approach. The framework in Ochoa-Hueso (2021), One Earth, may be useful for that.

Thank you for the excellent suggestion. We incorporated this null-model approach and rewired each observed microbial network 1000 times. We then calculated the modularity of each rewired network to create a chance distribution of the clustering of random networks, and calculated whether the observed networks were more, equally or less clustered than by chance (L517-521). All observed networks were significantly more densely clustered than would be expected based on chance indicating a distinct organisational pattern occurring in each network (L142-144; Fig S7).

L180. I found the results associated with this paragraph very complex to understand, in part because Fig 4 is not particularly informative/clear.

We revised the paragraph and particularly focussed on displaying the message of Figure 4 more clearly (L158-185). We also rewrote the caption of Figure 4 to improve its clarity.

L186. What do you mean?

The sentence was deleted.

L188-189. The meaning of this is unclear.

The sentence was deleted.

Figure 4. The meaning of this figure is unclear to me.

We revised the paragraph and particularly focussed on displaying the message of Figure 4 more clearly (L158-185). We also rewrote the caption of Figure 4 to improve its clarity.

L263. Was not it the other way around? Now I am confused.

No, this notion is correct. Abandoned arable soil was associated with destabilising properties both above- and belowground, not natural grassland soil.

L282-283. This is not necessarily linked with niche occupancy. It simply means that fungal and prokaryotic communities are less spatially structured in grassland soils, which may have to do with the destabilizing role of plant communities developed over these soils as compared to those growing on exarable land soils.

This is a good point, thank you. We deleted this interpretation from both the section on natural grassland soil as from the previous section on abandoned arable soil.

Reviewer #3 (Remarks to the Author):

This is a manuscript on microbial networks in abandoned and natural grasslands. Authors have made a clever experiment which allows them to link plant community stability and microbiomes. The manuscript is well written and complicated analyses presented in a simple way. Especially figure 6 is wonderful. The topic investigated is pretty novel and here many new aspects are presented. This will contribute to the plant literature although drawing causality from this data is not advised. I really like this manuscript, but do think certain aspects could be investigated with more statistical analysis and rigor. Especially the microbial side is pretty light while plant theory is well established. It is further a pity that microbes are analysed only once and that rhizosphere was not included in the analysis. Rhizosphere is the part of soil that would be in theory affected more by current plants while bulk soil more by previous plants (and does contain more dead microbes still having DNA). This is something that cannot be addressed here due to lack of sampling points and compartments.

Thank you very much! Please find our detailed responses to the raised issues below.

Statistical issues:

NMDS is not a suitable method to investigate differences between loadings (i.e. residence period, residence time). Here also more statistically robust methods should be used to evaluate the differences explained in community structure. This is pretty descriptive now. It would be important to see the statistics for the full NMDS and not per axis, for visual purposes it is ok to use them but this needs further interpretation with proper statistical tests.

Thank you for highlighting this issue. We were not aware that NMDS technically do not have axes scores and is more a visualisation technique. We re-analysed the plant community data using Detrended Correspondence Analysis (DCA). DCA calculates axes scores that have an interpretation, does not result in interdependence of data points resulting from multivariate analysis on dissimilarity matrices¹ and follows a unimodal distribution rather than a linear approach avoiding common horse shoe effects. We then redefined the interpretation of the compositional axes scores and recalculated compositional trends in time. We re-ran all structural equation models that previously incorporated the NMDS scores and re-calculated figure 4 summarising the relative importance of various pathways.

Correlating NMDS (that consist that parameter) with a parameter is not a neat way to analyse data.

We agree and changed the analyses. We now create a passive overlay of plant species traits on the DCA scores and test their significance using permutation tests (Fig S1, L442-446).

Using networks and clusters shows more or less the same results as would be captured with more traditional analysis. In general the ms is quite descriptive and light on statistics. However, the figures are clear. Interestingly, for microbes PcoA was used. This would be a suitable method also for plants. For microbes, it would have been good to know how different the communities were at the beginning and not only after 13 cycles. The soil chemistry was described in the beginning but we have no information on soil microbiome at the beginning of the experiment. This could be potentially one of the driving forces explaining some of the variation.

Thank you for the suggestion of using PCoA for the vegetation analysis. We have adapted our analysis of the vegetation (see above), although we now used DCA as it does not create interdependence of data points due to the use of a dissimilarity matrix for calculating multivariate scores like PCoA would¹. As the reviewer mentions, unfortunately, no microbial samples were taken at the start of the experiment in 2007. We have, however, incorporated differences in soil chemistry as a likely explanation in the different network structures between natural grassland and abandoned arable soil in the discussion, L325-352.

Other

points:

The results on past plant communities predicting soil microbial community are interesting. However, microbes were measured only once and they might fluctuate as well yearly. In the SEM link between

plants and microbes is made but it is unclear if these effects of especially past plants are direct or indirect (plants affecting other plants or soil abiotics). The effects are nicely shown but some links (plants on plants for example and between fungi and bacteria) are ignored.

We agree with the reviewer that it was not clear what the most likely underlying pathways were that shaped these past plant community pathways. We have clarified the differences in direct and indirect pathways in the new figure 5 and L188-204. In addition, we present the most consistently affected soil microbial groups by past plant community pathways, L237-259. In the discussion, we have added a discussion on the potential mechanistic pathways of past plant community pathways, L288-322.

Figure 5 is intriguing, it is not easy to understand as it is something between a network and a model. It does show clearly how complex the situation is and how all the parameters are connected.

We agree it is a difficult figure and therefore moved it to the supplements (Fig S10; see also comments reviewer 2).

Table 1 is quite dense in information and takes some moments to go through. Here more graphical representation would be advised.

Yes, we agree and deleted the table. We now present the most important pathways in the new Figure 5. All detailed information can, however, still be found in Tables S7 and S8.

It is interesting that fungi and bacteria would fully occupy separate soil niches. There must be large overlap for certain species while for others much less. If we talk about niches we should also acknowledge the scale we look at. For bacteria a whole soil sample is the size of an universe. We do not actually know where it is located in the soil (in water, in air, on an aggregate, inside an aggregate etc etc). Therefore it is tricky to discuss niche theory here.

We fully agree and deleted the sections on niches and niche differentiation from the discussion.

Figure 6 is wonderful!

Thank you very much!

Line 325: it would be helpful to use similar wording throughout. Here 'soil origin' is confusing.

We agree and now use 'soil origin' consistently throughout the whole manuscript.

Line 477: the bioinformatic analysis could be explained more clearly. How was Bray-curtis calculated on centerED log ratio? This is quite unusual way to analyse data.

Thank you for highlighting this issue. Following recommendations of Gloor et al.², we now use PCA to show the separation of soil microbial communities in Fig 2. We also adjusted to the typo in L491. In addition, we moved the section on the bioinformatics from the Supplementary Materials and Methods to the main section, L464-487.

Permanova is done but not reported?

Thank you, we added the PERMANOVA results that had disappeared from Fig 2.

As the putative life strategies of fungi and 'functions' of bacteria are included in the text, it would be very nice to know how authors derived these without digging deep into the supplementary materials. In supplementary materials also it is only said FungalTraits are used and nothing about confidence of the assignment. For bacteria nothing is said about functions yet in the text bacterial functions are discussed and appear suddenly in supplementary tables. Here either add references or use a source for all assignments.

The reviewer is quite right. We added a brief explanation on this approach to the result section in L208-214. We also added the confidence of FungalTraits assignment in the methods section (L479). As suggested by the reviewer, we also added references to Tables S2 and S3 to justify the putative

characteristics of the prokaryote network clusters. To further aid in the inference of the microbial networks, we calculated cluster-specific relative habitat specialisation indices and enrichment of the cluster in natural grassland or abandoned arable soil (Fig S8 and S9), L490-503 and L522-538.

In general, as microbes play an important role in the manuscript, it would be very important to balance also in mat&met between plant and microbiome information.

We added various microbial analyses to the manuscript and the materials and methods section: relative habitat specialisation indices, network cluster enrichment, OTU overlap between the two soils, comparison of clustering of the observed networks to a randomly rewired network. In addition, we moved the section on the bioinformatics from the Supplementary Materials and Methods to the main section to balance the plant and the microbial part, L464-487.

The definition of the clusters should be explained better as it is not really visible in figure S5

We agree and added a brief explanation on this approach to the result section in L208-214. A more detailed explanation can be found in the materials and methods, L522-538.

Interestingly for abandoned grasslands almost all clusters only contain species characterized as 'other' (Fig. S6). What could be a reason for this?

Thank you. We realised that the way we defined 'other' was misleading and recalculated this group for both the prokaryote and fungal taxonomy figures (Fig S5) and the fungal trait figures (Fig S6).

References

1. Legendre, P., Borcard, D. & Peres-Neto, P. R. Analyzing beta diversity: partitioning the spatial variation of community composition data. *Ecol Monogr* **75**, 435–450 (2005).
2. Gloor, G. B., Macklaim, J. M., Pawlowsky-Glahn, V. & Egozcue, J. J. Microbiome Datasets Are Compositional: And This Is Not Optional. *Front Microbiol* **8**, (2017).

Reviewers' Comments:

Reviewer #1:

Remarks to the Author:

Nice work

Reviewer #2:

Remarks to the Author:

This is the second time that I review this paper. Something that I did not say in my first review is that I very much like the experimental approach taken here, and so I would like to compliment the authors for the work done. I also agree with the type of questions that they are seeking to understand, and believe that this type of experiments are a great resource for that. However, I still find the paper quite defocused, most likely due to the complexity of describing so many plant, soil and microbial responses simultaneously. Importantly, I am sure the authors are correct in their interpretation, but I personally failed to visualize the main results as they were claimed. In other words, I could not see the different coupling/decoupling of fungal-prokaryote networks based on the figure provided, nor there was any metric supporting these claims. Moreover, the stability aspect is not adequately analysed either. For example, you could have analysed the stability of the plant communities in each pot from a temporal perspective (for example, using the inverse of the coefficient of variation), and then link this stability (based on diversity or biomass) to microbial network properties linked to coupling/decoupling. The introduction is highly focused on stability, but then there are no metrics of stability whatsoever reported in the paper. This is a major drawback of the paper as it stands in my opinion. Thus, I still partially concur with my previous comment that the introduction, results and discussion are decoupled, although some aspects have improved significantly as compared to the previous version.

P13-24. The abstract has some sentences that are grammatically incorrect, or that sound weird at best.

L27-28. It can also be argued that plants are a structuring force of microbial networks. Actually, this seems to be one of your main conclusions later on, then I wonder whether setting up this rationale is the best strategy for your paper.

L37. What do you mean by comprehensible structure?

L47. Two "via" in the same sentence.

L59. "Of indirect", space needed.

L60. Are these chemical changes induced by plants? I guess so, but the sentence is unclear.

L124. Is this a legacy effect due to greater fertility at the beginning? Well, the response seems to be yes, but I only found out later on when I read the discussion and methods. Would it not make sense to give a hint of this information earlier, for example when you briefly describe the experiment before the results?

L150-152. I cannot see this result. I see connecting lines between fungal and prokaryote clusters in both types of soils? What am I missing? Also, is there any metric that you can use to evaluate the degree of coupling/decoupling? I think that these results of greater diversity and stability of plants linked with greater decoupling is intriguing and makes complete sense, because if plants are in control, then microbes are not. And conversely, when plants are not in control, microbial networks are more likely to self-organize in networks that are highly spatially (and likely also temporally) structured. But, to be honest, I had a hard time visualizing this result.

Figure 4. I would like to see the model, and not only the relative contributions.

L192-199. Reading this paragraph, I wonder what the original goal of the paper was and whether the paper has drifted from its main goals. Where is stability in all this?

L366-367. This is a very interesting result.

Reviewer #3:

Remarks to the Author:

This is an interesting manuscript presenting work done in long-term trajectories of plant communities in two different soils (natural grassland and abandoned arable soil). The authors look into the stability of plant communities by analysing plant communities during 13 years and

networks of soil microbes at the end of the experiment. This is a significant contribution to the field of plant (community) science and increases our understanding on the community assembly. The authors have (based on suggestions of reviewers from previous round, including myself) improved the manuscript and now it presents a balanced level of information both on plants and on microbial networks. Also, the minor flaws that were in the methods have been corrected and everything explained in much more detail now making it clear. Overall, this will be an important contribution to the field and will for sure also spark discussion on the microbial networks.

Kind regards,
Emilia Hannula

REVIEWER COMMENTS

Reviewer #1 (Remarks to the Author):

Nice work

Thank you very much!

Reviewer #2 (Remarks to the Author):

This is the second time that I review this paper. Something that I did not say in my first review is that I very much like the experimental approach taken here, and so I would like to compliment the authors for the work done. I also agree with the type of questions that they are seeking to understand, and believe that this type of experiments are a great resource for that.

Thank you very much!

However, I still find the paper quite defocused, most likely due to the complexity of describing so many plant, soil and microbial responses simultaneously. Importantly, I am sure the authors are correct in their interpretation, but I personally failed to visualize the main results as they were claimed. In other words, I could not see the different coupling/decoupling of fungal-prokaryote networks based on the figure provided, nor there was any metric supporting these claims.

*We understand the reviewers comment and realise we interpreted this comment in the previous round of revisions incorrectly. Hence, the additional analyses we provided did not specifically test for the **decoupling** of fungal and prokaryote clusters, but solely for the occurrence of fungal and prokaryotes clusters itself. We have therefore expanded the previously added randomisation analyses to also include a calculation on the chance that coupling/decoupling of prokaryote and fungal clusters occur randomly. For this, we created 1000 random microbial networks for the prokaryotes and fungi and the two soil origins each. Each iteration, we tested for the occurrence of significant correlations between the prokaryote and fungal network clusters (L553-560; track changed document). We found that coupling of prokaryote and fungal clusters occurred significantly more often than would be expected based on chance for both soil origins (Fig S8). We added this conclusion to the results in L172-175.*

Moreover, the stability aspect is not adequately analysed either. For example, you could have analysed the stability of the plant communities in each pot from a temporal perspective (for example, using the inverse of the coefficient of variation), and then link this stability (based on diversity or biomass) to microbial network properties linked to coupling/decoupling. The introduction is highly focused on stability, but then there are no metrics of stability whatsoever reported in the paper. This is a major drawback of the paper as it stands in my opinion. Thus, I still partially concur with my previous comment that the introduction, results and discussion are decoupled, although some aspects have improved significantly as compared to the previous version.

Thank you, this is a great idea. We have performed the suggested analysis. We calculated temporal stability based on aboveground productivity of each plant community as temporal stability is most commonly calculated based on aboveground productivity (Hallett et al 2016 Methods Ecol Evol 7, 1146–1151). We found that temporal stability was indeed higher in the natural grassland communities than in abandoned arable communities (Fig 1D). We then sought a parameter describing coupling/decoupling of the prokaryote and fungal community within each sample/community. We found coupling/decoupling was well-described by the beta slope of the correlation between the relative reads summed per prokaryote clusters 1-9 and per fungal network clusters 1-9 within each sample/community. This correlation is valid as cluster assignment within the microbial networks is based on the order of prevalence of the ASVs in each cluster, hence the similarity in the patterns of the natural grassland prokaryote clusters and the abandoned arable prokaryote and fungal clusters visible in figure 3AB. It then follows that a positive correlation (positive beta slope) between prokaryote

clusters 1-9 and fungal clusters 1-9 per community indicates that the distribution of prokaryote and fungal reads among the most prevalent clusters in the networks is similar, i.e., prokaryote and fungal responses are coupled. A weak correlation/weak beta slope indicates that prokaryote and fungal reads are not similarly distributed among the most prevalent network clusters, and thus prokaryote and fungal responses are decoupled. In a few cases, we found a weak negative correlation, which indicates that the distribution of prokaryote and fungal reads over the most prevalent clusters was nearing an opposite pattern, i.e., a very strong decoupling in the responses of prokaryote and fungi. We added this calculation to the methods in L561-573 and the caption of figure 4.

We correlated the beta slope parameter to the temporal stability of the plant communities and found a significant negative relation. As expected, this relation indicates that plant communities with a high temporal stability were decoupled in prokaryote and fungal network responses, while plant communities with a low temporal stability were coupled in prokaryote and fungal network responses. We added these findings to the result section in L176-179 and Fig 4, and the discussion L297-299.

P13-24. The abstract has some sentences that are grammatically incorrect, or that sound weird at best.

We went through the abstract again and corrected any sentences that may have been perceived as weird (L13-28).

L27-28. It can also be argued that plants are a structuring force of microbial networks. Actually, this seems to be one of your main conclusions later on, then I wonder whether setting up this rationale is the best strategy for your paper.

Certainly, and this is why we introduce these plant-microbial networks as interactive networks in the previous sentence. Throughout the manuscript, we are cautious as to pinpoint the exact direction of these interactions. In the previous revision, we reworded any 'effects' to 'interactions', 'associations' and 'pathways' to avoid overinterpretation of the interactions between plants and microbiota.

L37. What do you mean by comprehensible structure?

We mean that the network structures are understandable, predictable, interpretable, coherent. We changed 'comprehensible' into 'coherent' to clarify (L41).

L47. Two "via" in the same sentence.

Thank you. We adjusted the sentence to have only one 'via' in it (L51-52).

L59. "Of indirect", space needed.

Thank you. We added a space (L54).

L60. Are these chemical changes induced by plants? I guess so, but the sentence is unclear.

Not necessarily just by plants. Microbiota are the engines of soil N and C cycling and therewith induce soil chemical changes along with plants. We adjusted the sentence (L64).

L124. Is this a legacy effect due to greater fertility at the beginning? Well, the response seems to be yes, but I only found out later on when I read the discussion and methods. Would it not make sense to give a hint of this information earlier, for example when you briefly describe the experiment before the results?

Thank you for the suggestion. We moved the sentence on these soil differences from the method section to the introduction where we introduce the experiment (L73-75).

L150-152. I cannot see this result. I see connecting lines between fungal and prokaryote clusters in both types of soils? What am I missing? Also, is there any metric that you can use to evaluate the degree of coupling/decoupling? I think that these results of greater diversity and stability of plants

linked with greater decoupling is intriguing and makes complete sense, because if plants are in control, then microbes are not. And conversely, when plants are not in control, microbial networks are more likely to self-organize in networks that are highly spatially (and likely also temporally) structured. But, to be honest, I had a hard time visualizing this result.

The connecting lines are significant correlations between the microbial clusters. These indicate whether these clusters respond similarly across the 30 communities of each soil (positive correlation) or opposite (negative correlation) or have nothing in common (no significant correlation) (see caption Figure 3). L158-163 introduces the dominant clusters in the two soils and the remarkable differences within this. We calculated and added the average relative abundance of the read counts that were held by the dominant clusters to demonstrate the dominance of these clusters more quantitatively. L164-170 states that only a single correlation between these dominant prokaryote and fungal clusters occurred in natural grassland soil (prokaryote cluster 3 and fungal cluster 6), while many more were present in abandoned arable soil (prokaryote and fungal clusters 1-3). We now point out the specific clusters that correlated in the text. Additionally, as suggested by the reviewer, we calculated aboveground temporal stability and correlated this to coupling/decoupling of the prokaryote and fungal soil community (see previous comment). We added these results in L176-179, which should clarify the message even further.

Figure 4. I would like to see the model, and not only the relative contributions.

Unfortunately, the model itself is completely incomprehensible when not summarised in relative contributions. All pathways (in absolute form) can, however, be found in Fig S11 and S12. We added a reference to these supplementary figures in the caption of Figure 5.

L192-199. Reading this paragraph, I wonder what the original goal of the paper was and whether the paper has drifted from its main goals. Where is stability in all this?

This and the following section dive deeper into the explicit pathways that shape the decoupling/coupling of prokaryote and fungal networks. This relates directly to stability as stability and prokaryote/fungal coupling/decoupling were significantly associated. We clarified this context in L213-214.

L366-367. This is a very interesting result.

Thank you!

Reviewer #3 (Remarks to the Author):

This is an interesting manuscript presenting work done in long-term trajectories of plant communities in two different soils (natural grassland and abandoned arable soil). The authors look into the stability of plant communities by analysing plant communities during 13 years and networks of soil microbes at the end of the experiment. This is a significant contribution to the field of plant (community) science and increases our understanding on the community assembly. The authors have (based on suggestions of reviewers from previous round, including myself) improved the manuscript and now it presents a balanced level of information both on plants and on microbial networks. Also, the minor flaws that were in the methods have been corrected and everything explained in much more detail now making it clear. Overall, this will be an important contribution to the field and will for sure also spark discussion on the microbial networks.

Kind regards,

Emilia Hannula

Thank you very much!

Reviewers' Comments:

Reviewer #2:

Remarks to the Author:

I am very thankful to the authors for the way in which they handled and responded to my comments. This is now the paper that I expected when I first read it, and I wish to compliment the authors for the work done. The relationship between coupling and stability is a very exciting result, and this is now going to be a very nice contribution!

I only have a relatively minor, yet relevant comment. The word coupling is completely missing from the introduction, but this is a very relevant keyword in the manuscript, so I think it should be present. I thus suggest rewriting the first sentence of the first paragraph to accommodate this word. You could open the paper with something like: "Plants and soil microbial communities are highly coupled through complex interactive networks". Alternatively, you can introduce the word elsewhere in the introduction. Moreover, you could very briefly even expand on what is coupling and what it means. This paper (Ochoa-Hueso et al., 2021, One Earth, <https://www.sciencedirect.com/science/article/pii/S2590332221003535>), of which I am the lead author, may help you to provide such a framework, both in the Introduction, and later on in the Methods when you describe decoupling based on a null modelling approach. I think it is important to cite this paper because this would allow you to link your study with a growing body of literature focused on how ecosystems are coupled, and on what this coupling means in terms of functioning, stability, etc. Actually, it would be a perfect citation for your opening sentence, which is currently lacking one, despite being one of the most important sentences in your paper.

Again, congratulations on the work done!

Best,

Raúl Ochoa-Hueso

Response to reviewer comments

REVIEWERS' COMMENTS

Reviewer #2 (Remarks to the Author):

I am very thankful to the authors for the way in which they handled and responded to my comments. This is now the paper that I expected when I first read it, and I wish to compliment the authors for the work done. The relationship between coupling and stability is a very exciting result, and this is now going to be a very nice contribution!

I only have a relatively minor, yet relevant comment. The word coupling is completely missing from the introduction, but this is a very relevant keyword in the manuscript, so I think it should be present. I thus suggest rewriting the first sentence of the first paragraph to accommodate this word. You could open the paper with something like: "Plants and soil microbial communities are highly coupled through complex interactive networks". Alternatively, you can introduce the word elsewhere in the introduction. Moreover, you could very briefly even expand on what is coupling and what it means. This paper (Ochoa-Hueso et al., 2021, One Earth, <https://www.sciencedirect.com/science/article/pii/S2590332221003535>), of which I am the lead author, may help you to provide such a framework, both in the Introduction, and later on in the Methods when you describe decoupling based on a null modelling approach. I think it is important to cite this paper because this would allow you to link your study with a growing body of literature focused on how ecosystems are coupled, and on what this coupling means in terms of functioning, stability, etc. Actually, it would be a perfect citation for your opening sentence, which is currently lacking one, despite being one of the most important sentences in your paper.

Again, congratulations on the work done!

Best,

Raúl Ochoa-Hueso

Thank you for your kind words. The reviewer raises a valid point, we indeed did not include ecosystem coupling/decoupling in our introduction. We carefully read the publication written by the reviewer and we acknowledge the reviewer's perspective on ecosystem coupling and appreciate their work in highlighting the concept. However, we would like to clarify that our approach to coupling/decoupling in our paper differs from the reviewer's concept. Our focus is on network science and cluster formation rather than overall ecosystem connectivity according to which Ochoa-Hueso et al. define coupling/decoupling. Specifically, we define coupling/decoupling of prokaryote and fungal communities alone, not of the ecosystem as a whole. This is because connectivity may result in vulnerability depending on where in the network/ecosystem connections occur (Barabási & Pósfai, 2016, Network Science). Our results can therefore only be understood by placing coupling/decoupling of prokaryotes and fungi in the concept of network cluster formation, which is not per se related to overall ecosystem connectivity/the concepts highlighted by Ochoa-Hueso et al (see Figure 7).

To introduce the coupling/decoupling of prokaryotes and fungi in the introduction, we extended our introduction on cluster formation in L38-40 (tracked change document). Here we now explicitly place coupling/decoupling within the context of cluster formation to avoid confusion with approaches focussing on overall ecosystem connectivity. Additionally, we now also explicitly incorporated network science in L44-L47 to avoid confusion on the background of concepts in the study.